# Structural basis of myelin-associated glycoprotein adhesion and signalling

Matti F. Pronker[1], Suzanne Lemstra[2], Joost Snijder[3], Albert J.R. Heck[3], Dominique M.E. Thies-Weesie[4], R. Jeroen Pasterkamp[2] & Bert J.C. Janssen[1]

Myelin-associated glycoprotein (MAG) is a myelin-expressed cell-adhesion and bi-directional signalling molecule. MAG maintains the myelin–axon spacing by interacting with specific neuronal glycolipids (gangliosides), inhibits axon regeneration and controls myelin formation. The mechanisms underlying MAG adhesion and signalling are unresolved. We present crystal structures of the MAG full ectodomain, which reveal an extended conformation of five Ig domains and a homodimeric arrangement involving membrane-proximal domains Ig4 and Ig5. MAG-oligosaccharide complex structures and biophysical assays show how MAG engages axonal gangliosides at domain Ig1. Two post-translational modifications were identified—N-linked glycosylation at the dimerization interface and tryptophan C-mannosylation proximal to the ganglioside binding site—that appear to have regulatory functions. Structure-guided mutations and neurite outgrowth assays demonstrate MAG dimerization and carbohydrate recognition are essential for its regeneration-inhibiting properties. The combination of *trans* ganglioside binding and *cis* homodimerization explains how MAG maintains the myelin–axon spacing and provides a mechanism for MAG-mediated bi-directional signalling.

[1] Crystal and Structural Chemistry, Bijvoet Center for Biomolecular Research, Department of Chemistry, Faculty of Science, Utrecht University, Padualaan 8, 3584 CH Utrecht, The Netherlands. [2] Department for Translational Neuroscience, Brain Center Rudolf Magnus, University Medical Center Utrecht, Universiteitsweg 100, 3584 CG Utrecht, The Netherlands. [3] Biomolecular Mass Spectrometry and Proteomics, Bijvoet Center for Biomolecular Research and Utrecht Institute for Pharmaceutical Sciences, Department of Chemistry and Department of Pharmaceutical Sciences, Faculty of Science, Utrecht University, Padualaan 8, 3584 CH Utrecht, The Netherlands. [4] Van't Hoff Laboratory for Physical and Colloid Chemistry, Debye Institute of Nanomaterials Science, Department of Chemistry, Faculty of Science, Utrecht University, Padualaan 8, 3584 CH Utrecht, The Netherlands. Correspondence and requests for materials should be addressed to B.J.C.J. (email: b.j.c.janssen@uu.nl).

Myelination of axons enables enhanced conductance velocity in both the central and peripheral nervous system (CNS and PNS) of vertebrates. It also provides electrical insulation and a decrease of the capacitance, as well as physical protection and metabolic support of long axons[1]. Myelin-associated glycoprotein (MAG) adhesion and signalling at the myelin–axon interface regulates the formation and maintenance of myelinated axons, thus playing an important role in the development and function of the nervous system[2,3]. Aberrant MAG function, for example from mutations that likely cause misfolding, or anti-MAG autoimmunity, has been associated with demyelination and neurodegenerative disorders, such as corticospinal motor neuron disease also known as hereditary spastic paraplegias[4], Pelizaeus–Merzbacher disease-like disorder[5], demyelinating anti-MAG peripheral neuropathy[6,7] and multiple sclerosis[2,8].

MAG is a type 1 single-pass transmembrane protein expressed on myelinating oligodendrocytes in the CNS and Schwann cells in the PNS[2,3]. MAG is the fifth highest expressed protein in myelin of the CNS[9]. It is highly enriched at the innermost (adaxonal) myelin membrane along the internode, where it contacts the axon. MAG is also found on other myelin structures, such as the mesaxon, Schmidt-Lanterman incisures and paranodal loops[2,3]. MAG adhesion maintains the myelin–axon spacing (periaxonal diameter) by interacting with specific neuronal gangliosides (glycolipids), such as the major brain gangliosides GT1b and GD1a (refs 10–13). More recently, the Nectin-like (Necl) proteins 1 and 4 have also been found to contribute to myelin–axon adhesion along the internode[14,15], although they are expressed less than MAG in mature myelin[9] and knockout of Necl4 does not affect myelination[16].

MAG, also known as Siglec4a, is evolutionarily the oldest member of the Siglec family[17]. Unlike all other Siglecs, MAG plays no role in the immune system and is exclusively expressed in the nervous system[17]. On the basis of the primary sequence its extracellular region is predicted to consist of five Ig domains; an N-terminal V-type Ig domain that is typical for Siglecs and four C2-type Ig domains. This is followed by a single membrane-spanning helix and an intracellular region predicted to be unstructured and of different length for two MAG isoforms, L-MAG and S-MAG. Like other Siglecs, MAG recognizes sialic acid groups and the specificity of MAG has been established to be Neu5Ac-α2,3-Gal-β1,3-GalNAc (ref. 18). This trisaccharide is part of several neuronal gangliosides, most notably the major brain gangliosides GT1b and GD1a, but also GM1b, GT1β and GQ1bα. MAG bridges the periaxonal space by interacting with these axonal gangliosides in *trans* via the canonical Siglec site at a conserved arginine (R118 in MAG) in the N-terminal domain[19,20].

MAG signalling is bidirectional, engaging in both axon-to-myelin as well as myelin-to-axon signalling. MAG has been extensively studied as one of three classic myelin-associated inhibitors of central nervous system regeneration, the other ligands being Nogo66 and oligodendrocyte myelin glycoprotein[2,3]. MAG inhibits neurite outgrowth and collapses axonal growth cones in a sialic acid binding-dependent manner. It does so as full-length transmembrane[20,21], but also as a proteolytically shed and soluble form called dMAG[22]. As a receptor, MAG controls myelin formation and integrity. How MAG transduces the extracellular signal into the myelinating cell is not well understood, but it has been shown that the cytosolic domain of the L-MAG isoform binds to the cytoplasmic non-receptor tyrosine kinase Fyn[23] and that antibody-induced crosslinking of L-MAG triggers its localization to lipid rafts[24] and activates Fyn in oligodendrocytes[23]. This activation of Fyn is essential for the initiation of myelination[25]. In contrast, the shorter MAG isoform

S-MAG binds to zinc and microtubules and this is postulated to have a structural function in mature myelin[26,27].

From earlier rotary-shadowed electron microscopy (EM) and sedimentation velocity analytical ultracentrifugation (AUC) studies it was hypothesized that the extracellular segment of MAG has a back-folded Ig-horseshoe type structure, but the estimated maximum dimensions of 8.8 and 18.5 nm determined by AUC and EM, respectively, deviate substantially[28,29]. In the absence of any high-resolution structural data on MAG or its interaction with ganglioside ligands, the conformation of the five Ig domains, the extracellular specificity-determining parameters and the mechanisms underlying MAG adhesion and bi-directional signalling are unresolved. Using a combination of structural, biophysical and cellular techniques, we provide the structural basis of MAG-mediated adhesion and identify a dimerization-dependent mechanism that explains how MAG regulates axon-to-myelin and myelin-to-axon signalling, and controls myelin–axon spacing.

## Results

**MAG has an extended conformation**. We determined crystal structures of the full extracellular segment of mouse MAG (MAG$_{1-5}$) in two different crystal forms that diffracted to a maximum resolution of 3.8 and 4.3 Å. These crystals were obtained by enzymatic deglycosylation of MAG$_{1-5}$ or reductive lysine methylation of glycosylated MAG$_{1-5}$ (see 'Methods' section). In addition, crystals of a shorter construct, consisting of the three N-terminal domains (MAG$_{1-3}$), diffracted to a maximum resolution of 2.1 Å. The structures were solved by molecular replacement with individual Ig domains from homologous proteins. The exceptionally high-solvent content of the two MAG$_{1-5}$ crystal forms (91 and 85%, Supplementary Fig. 1) aided in obtaining phases of sufficient quality for initial model building (see also Table 1 and 'Methods' section for details).

In all three crystal forms MAG has an extended collinear conformation (Fig. 1). Only consecutive Ig domains interact with each other via hydrophobic interfaces (buried-surface area ranging from 243 to 690 Å$^2$) and short inter-domain linkers of up to two residues (Fig. 1a). Domains Ig1 and Ig2 form the largest interface in which the Ig2 loops at the N-terminal 'head' side interact with the A2-B (Ig domain β–strand numbering) side of Ig1 (Fig. 1a). The three other inter-domain interfaces are exclusively formed in a head-to-tail manner involving loops at the N-terminal 'head' and C-terminal 'tail' side of the Ig domains (Fig. 1a). As predicted from the primary sequence the N-terminal Ig1 domain of MAG has a V-type Ig fold like other Siglec family members[30–32] and domains Ig3 and Ig4 are of the C2 type. Domains Ig2 and Ig5, however, have a C1-type Ig fold, contrary to the predicted C2-fold (Supplementary Fig. 2). The three crystal structures of MAG are similar to each other with only small differences within the domains (r.m.s.d. ranging from 0.93 to 2.13 Å) and inter-domain angle rotation differences ranging from 3.4 to 17.4°, the largest difference is in the domain Ig2–Ig3 angle (Fig. 1b). The combination of hydrophobic inter-domain interfaces, the lack of flexible linker residues and a previously predicted inter-domain disulfide between Ig1 and Ig2 (C37–C165)[33] explains the limited inter-domain flexibility observed between the three different crystal forms.

**MAG is post-translationally modified**. The structures reveal MAG is post-translationally modified at several sites. MAG contains seven disulfides, five of which are canonical for Ig domains. Cysteines 37 and 165 form an inter-domain disulfide between Ig1 and Ig2, and cysteines 421 and 430 form an

**Table 1 | Data collection and refinement statistics.**

| | MAG$_{1-3}$ unliganded | MAG$_{1-3}$ ligand bound | MAG$_{1-5}$ deglycosylated | MAG$_{1-5}$ lysine-methylated |
|---|---|---|---|---|
| *Data collection* | | | | |
| Space group | P1 | P1 | P3$_2$2 | P6$_5$22 |
| Cell dimensions | | | | |
| $a$, $b$, $c$ (Å) | 43.06, 60.4, 79.22 | 43.61, 60.12, 79.47 | 278.9, 278.9, 62.52 | 101.2, 101.2, 687.5 |
| $\alpha$, $\beta$, $\gamma$ (°) | 72.70, 86,71, 83.01 | 71.86, 86.51, 82.95 | 90, 90, 120 | 90, 90, 120 |
| Resolution (Å) | 42.73–2.12 (2.19–2.12) | 56.79–2.30 (2.38–2.30) | 69.72–3.80 (4.03–3.80) | 114.62–4.30 (4.81–4.30) |
| $R_{sym}$ or $R_{merge}$ | 0.064 (1.118) | 0.157 (0.981) | 0.234 (1.683) | 0.115 (3.937) |
| Mean $I/\sigma I$ | 8.8 (0.9) | 5.9 (1.6) | 9.2 (1.6) | 15.6 (1.3) |
| CC$_{1/2}$ | 0.998 (0.558) | 0.985 (0.565) | 0.997 (0.565) | 0.998 (0.242) |
| Completeness (%) | 96.2 (95.7) | 97.5 (95.7) | 100.0 (100.0) | 100.0 (100.0) |
| Redundancy | 3.6 (3.5) | 4.5 (4.3) | 9.6 (9.7) | 35.7 (36.9) |
| | | | | |
| *Refinement* | | | | |
| Resolution (Å) | 43–2.1 | 57–2.3 | 70–3.8 | 115–4.3 |
| No. reflections | 42,931 | 42,931 | 35,257 | 15,430 |
| $R_{work}$/$R_{free}$ | 0.224/0.262 | 0.224/0.254 | 0.203/0.230 | 0.266/0.282 |
| No. atoms | | | | |
| Protein | 5,054 | 4,977 | 3,841 | 3,942 |
| Ligand/ion | 54 | 84 | 46 | |
| Water | 54 | 34 | | |
| Average $B$-factors | | | | |
| Protein | 71.1 | 70.9 | 144.2 | 414.5 |
| Ligand/ion | 106.8 | 102.5 | 129.4 | |
| Water | 60.8 | 59.7 | | |
| r.m.s. deviations | | | | |
| Bond lengths (Å) | 0.003 | 0.002 | 0.006 | 0.008 |
| Bond angles (°) | 0.700 | 0.583 | 1.218 | 1.384 |
| Molprobity score | 1.69 | 1.49 | 2.38 | 2.36 |
| Molprobity percentile | 94th | 99th | 99th | 99th |

r.m.s, root mean square.
*Each data set was collected from a single crystal.

additional intra-domain disulfide in Ig5, as shown previously[33] (Supplementary Fig. 3). In addition, MAG carries N-linked glycans and previously eight N-linked glycosylation sites were determined in human MAG by mass spectrometry analysis[34]. We observe glycan electron density for five of those equivalent sites in the mouse MAG structures (on asparagine 99, 223, 246, 315 and 406; Supplementary Fig. 4). The differences in these observations arise from one N-linked glycosylation site that is not conserved (N106 in human MAG is a threonine in mouse) and from poorly resolved electron density for the other two sites (on N450 and N454) that are situated in a flexible loop. One additional N-linked glycan is revealed by clear electron density on N332, in contrast to the previous study that did not find this residue in a glycopeptide analysis[34]. Electron density at the N332-linked glycan suggest it is fucosylated in our recombinantly produced MAG (Supplementary Fig. 5). In native mass spectrometry experiments of intact recombinant MAG we also observed internal mass shifts of +147 Da on MAG monomers (Supplementary Fig. 6), further indicating fucosylation as a post-translational modification. Thus the combination of our data and that of others[34] indicates that mouse MAG has eight N-linked glycosylation sites (on N99, N223, N246, N315, N323, N406, N450 and N454).

In addition, MAG carries a tryptophan C-mannosylation on W22. In all three crystal forms, electron density proximal to the side chain of W22 suggests this residue is C-mannosylated (Supplementary Fig. 6). Indeed, analysis of the MAG primary sequence reveals that this tryptophan is part of the canonical WxxW motif (W22 is the first tryptophan) for C-mannosylation, a rare post-translational modification present on several secreted proteins[35]. The W22-attached α-D-mannopyranosyl group has an unusual ring-flipped $^{1}C_4$ chair conformation in the structures

(regular mannose is $^{4}C_1$, Supplementary Fig. 6). This ring-flipped conformation is in agreement with previous nuclear magnetic resonance studies on mannosyl-tryptophan[36] and can be explained by the preference of the bulky tryptophan, covalently attached to the C1 of the mannose, to be in the equatorial position. We confirmed the C-mannosylation of W22 by in-gel trypsin digestion of MAG$_{1-5}$ followed by liquid chromatography-mass spectroscopy (LC-MS/MS) analysis of the (glyco)peptide fragments (Supplementary Fig. 7). In addition, native mass spectrometry of wild type and mutated MAG$_{1-5}$, in which the second tryptophan of the C-mannosylation motif is substituted for glutamine (W25Q, resulting in WxxQ), showed a mass shift of −221 Da in accordance with a loss of C-mannosylation and confirming the importance of the W̲xxW motif (Supplementary Fig. 6).

**MAG$_{1-5}$ crystal structures reveal a dimeric arrangement.** MAG$_{1-5}$ forms a symmetry-related dimer at a crystallographic two-fold rotation axis in both MAG$_{1-5}$ crystal forms. They share the same interface (Fig. 2a) on domains Ig4 and Ig5, which buries a surface area of 2,037 Å$^2$. The CC′FG face of Ig4 binds to the ABDE face of Ig5 of the symmetry-related molecule and vice versa, thus forming two equivalent hemi-interfaces. The interface is mostly hydrophobic apart from the negatively charged E395 in Ig4, with hydrophilic residues lining the edges of the interface (Fig. 2b).

To validate the interface we generated two interface mutations based on the structures, that we predicted to either disrupt dimerization (I473 to E) or enhance it (N406 to Q). The hydrophobic I473 in the middle of the hydrophobic interface of Ig5 was mutated to a negatively charged glutamate, to ensure disturbance of the hydrophobic effect as well as introducing

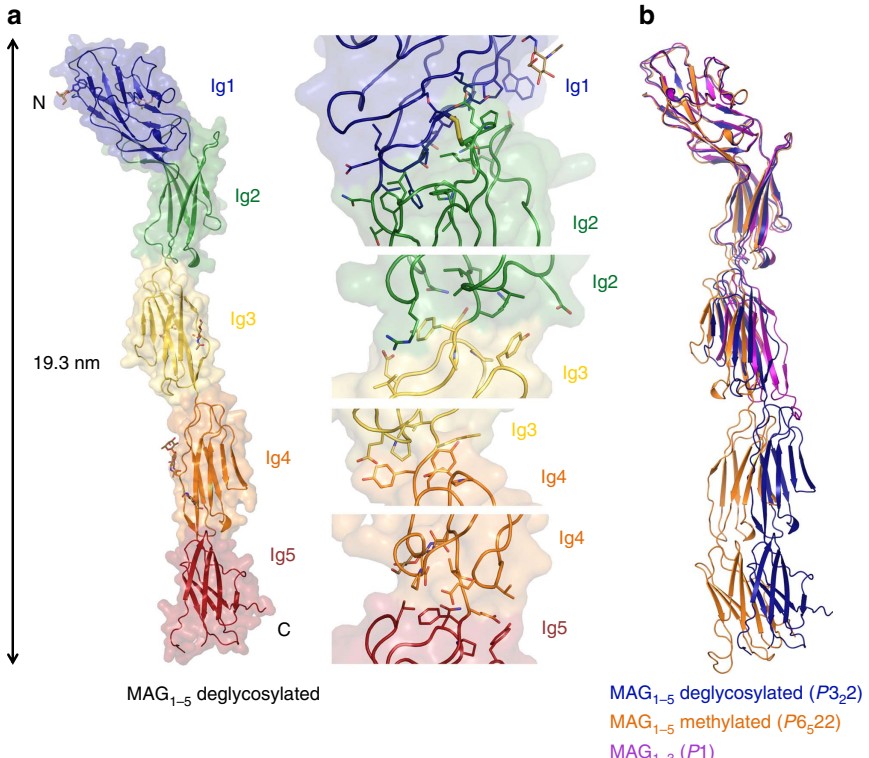

**Figure 1 | Crystal structures of MAG reveal an extended conformation with limited inter-domain flexibility. (a)** Crystal structure of deglycosylated $MAG_{1-5}$ coloured by domain, glycosylation sites indicated in stick representation and N- and C-termini indicated with N and C (left panel). Close-up views of the inter-domain interfaces with sidechains of interacting residues and the inter-domain disulfide between Ig1 and Ig2 shown in stick representation (right panel). **(b)** Superposition, based on domain Ig1, of the three crystal structures of MAG; deglycosylated $MAG_{1-5}$ (blue), lysine-methylated $MAG_{1-5}$ (orange) and $MAG_{1-3}$ (purple), spacegroups indicated in brackets.

electrostatic repulsion with the opposing E395 in Ig4 (Fig. 2c). N406 carries an N-linked glycan and in the glycosylated lysine-methylated crystals of $MAG_{1-5}$, the density for this glycan suggests that it sterically clashes with its symmetry partner in the dimer (Fig. 2d). We hypothesized that a dimer with increased affinity would form, were this glycan absent. Remarkably, the N406Q mutant that lacks this glycan was the only MAG construct that showed a distinct dimer peak in size exclusion chromatography (SEC; Supplementary Fig. 8). The other glycans are not expected to interfere with dimerization, also not when considering myelin-specific N-linked glycans[37]. We confirmed that MAG dimerizes in solution via the Ig4–Ig5 interface by analysing glycosylated and deglycosylated forms of $MAG_{1-5}$, $MAG_{1-3}$ (that lacks the dimerization domains) and the aforementioned mutants in small angle X-ray scattering (SAXS) and AUC experiments.

**SAXS confirms dimerization interface.** As predicted, SAXS analysis indicated disruption of dimerization for $MAG_{1-3}$ and $MAG_{1-5}$ I473E, whereas dimerization is enhanced for $MAG_{1-5}$ N406Q and deglycosylated $MAG_{1-5}$ compared with wt $MAG_{1-5}$. The molecular mass ($M_m$) based on the extrapolated intensity at zero scattering angle ($I_0$, scaled for concentration to bovine serum albumin (BSA)), the radius of gyration ($R_g$), the maximum interatomic distance ($D_{max}$) and the Porod volume all show these trends (Fig. 2, Supplementary Fig. 9 and Table 2). The data show that at similar concentrations, $MAG_{1-5}$ I473E appears smaller than wt $MAG_{1-5}$ and $MAG_{1-5}$ N406Q appears larger than wt $MAG_{1-5}$. This can be attributed to a shift in the monomer–dimer equilibrium; $MAG_{1-5}$ I473E has a lesser and $MAG_{1-5}$ N406Q a greater propensity to dimerize compared with wt $MAG_{1-5}$.

Furthermore, the $MAG_{1-5}$ I473E pair distance distribution function P(r) and derived $D_{max}$, the *ab initio* models as well as the Kratky plots confirm that MAG has an extended conformation and behaves as a semi-rigid rod in solution (Fig. 2f,g,i). Whereas the SAXS data for glycosylated $MAG_{1-3}$ and $MAG_{1-5}$ I473E fit best to scattering curves calculated from single chains of the crystal structures of $MAG_{1-3}$ and $MAG_{1-5}$, respectively ($\chi = 2.87$ and 2.95, Supplementary Fig. 10), the glycosylated $MAG_{1-5}$ N406Q SAXS data fits best to scattering curves calculated from the dimer structure ($\chi = 3.45$, Supplementary Fig. 10). Both glycosylated and deglycosylated $MAG_{1-5}$ wt SAXS data fit best to scattering curves calculated from a combination of monomeric and dimeric crystal structures ($\chi = 4.05$ for glycosylated and 3.72 for deglycosylated $MAG_{1-5}$, Fig. 2j and Supplementary Fig. 11). Furthermore, *ab initio* models based on the SAXS data of deglycosylated $MAG_{1-5}$ I473E and $MAG_{1-5}$ N406Q agree remarkably well with the crystal structures of the monomer and dimer of $MAG_{1-5}$, respectively (Fig. 2g, Supplementary Fig. 12, $\chi^2$ of the model-to-data fit are 1.05 and 1.33 for $MAG_{1-5}$ I473E and $MAG_{1-5}$ N406Q, respectively). These data confirm MAG dimerizes via domains Ig4 and Ig5 and that the $MAG_{1-5}$ chains have an extended and relatively rigid conformation.

**The $MAG_{1-5}$ dimer is weak in solution with a $K_d$ of $3.8 \times 10^2$ µM.** To quantify the affinity of MAG dimerization in solution, we performed sedimentation equilibrium AUC (SE-AUC) experiments for wt $MAG_{1-5}$, deglycosylated wt $MAG_{1-5}$, $MAG_{1-3}$, $MAG_{1-5}$ I473E and $MAG_{1-5}$ N406Q constructs. For each sample a global analysis was performed for different concentrations, centrifugal speeds and wavelengths together (Supplementary Figs 13–16), except for $MAG_{1-5}$ N406, which appeared to suffer

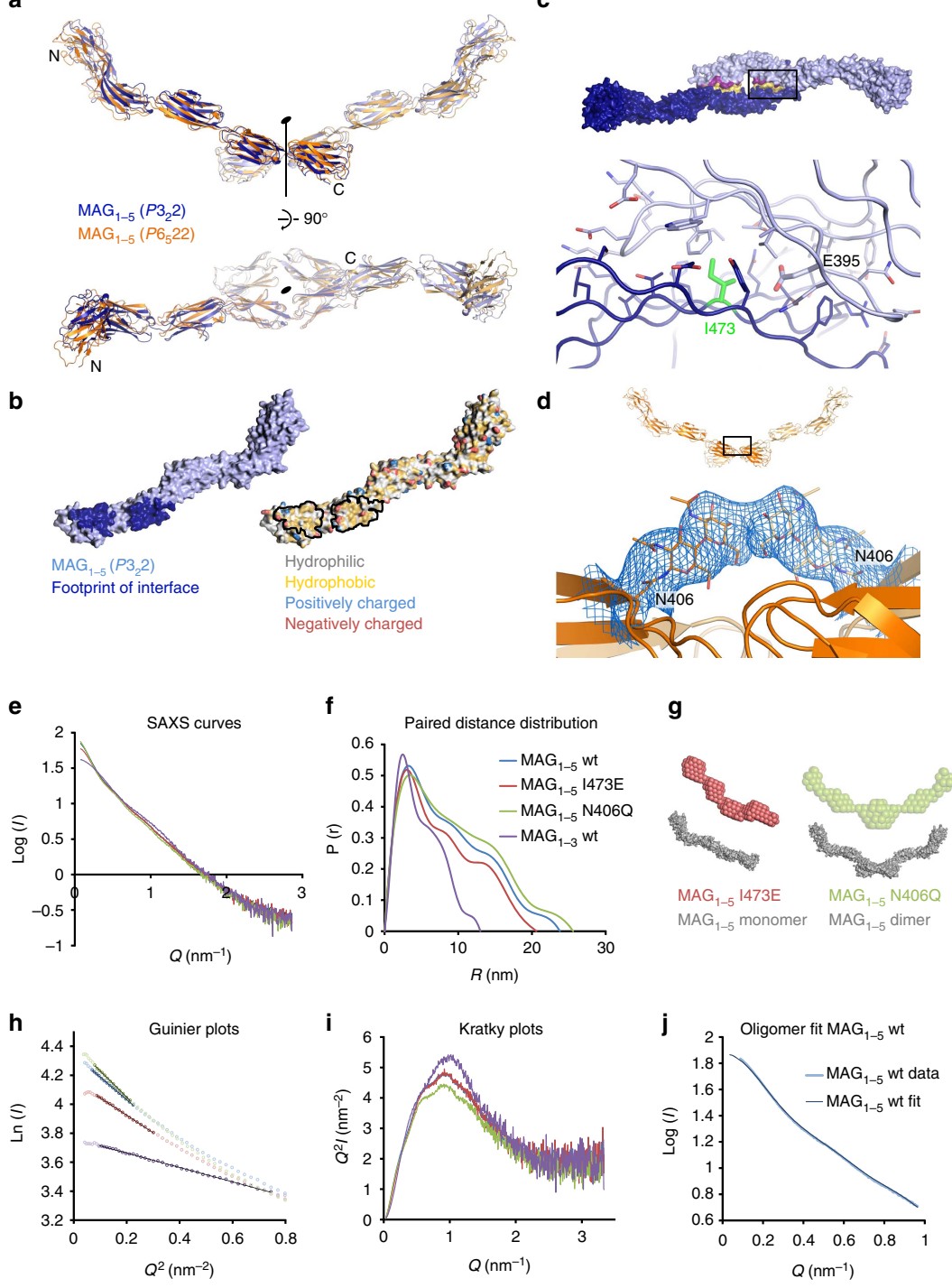

**Figure 2 | MAG forms dimers via domains Ig4 and Ig5. (a)** Superposition of the crystallographic dimers observed in the crystals of $MAG_{1-5}$ deglycosylated (blue) and $MAG_{1-5}$ lysine-methylated (orange). The two-fold axis is indicated by a black line with an ellipse on top, N- and C-termini indicated with N and C for one monomer. **(b)** The dimer is formed by two equivalent symmetry-related hemi-interfaces, which are mostly hydrophobic (yellow) with hydrophilic (grey), positively charged (blue) and negatively charged (red) residues lining the edges. **(c)** Zoom of the interface in deglycosylated $MAG_{1-5}$, indicating isoleucine 473, which was mutated to glutamate to disturb the interface. **(d)** $2F_o - F_c$ electron density at a contour level of 1.0 $\sigma$ of the N-linked glycan on N406 in the lysine-methylated $MAG_{1-5}$ crystal structure, showing the steric hindrance with its symmetry mate. Glycosylation at this site was prevented by mutating N406 to glutamine to obtain a MAG variant with enhanced dimerization properties. **(e)** SAXS Log I versus Q plots for glycosylated MAG variants: $MAG_{1-5}$ wt (blue), $MAG_{1-5}$ I473E (red), $MAG_{1-5}$ N406Q (green) and $MAG_{1-3}$ (purple), same colouring used in (**e–j**). **(f)** Paired distance distribution functions of the same MAG variants as in (**e**). **(g)** DAMMIF *ab initio* modelling for deglycosylated $MAG_{1-5}$ I473E (red) and $MAG_{1-5}$ N406Q (green), showing remarkable similarity to the crystal structures of monomeric MAG (left, grey) and the crystallographic dimer (right, grey), respectively (see Supplementary Fig. 13 for fits). **(h)** Guinier plots of the glycosylated MAG variants show the same size-related trends as the paired distance distribution function in (**f**). **(i)** Kratky plots for the different glycosylated MAG variants confirm that MAG behaves as a semi-rigid rod in solution and not as flexible beads-on-a-string. **(j)** OLIGOMER fit of the MAG monomer and crystallographic dimer to the glycosylated $MAG_{1-5}$ wt data, using the glycosylated and lysine-methylated crystal structures. OLIGOMER gives a 0.72:0.28 (monomer:dimer) ratio at a MAG concentration of 52 µM.

**Table 2 | SAXS data collection and parameters.**

| Sample | $M_m$ (kDa) | Concentration ($\mu$M) | Temperature (K) | $R_g$ (nm) | $M_m$ based on $I_0$ (kDa) | $D_{max}$ (nm) | Porod volume (nm$^3$) | SASBDB accession code |
|---|---|---|---|---|---|---|---|---|
| MAG$_{1-5}$ wt glycosylated | 64.9 | 52.2 | 293 | 6.8 | 76.8 | 23.8 | 153.8 | SASDB55 |
| MAG$_{1-5}$ I473E glycosylated | 64.9 | 46.2 | 293 | 6.0 | 64.4 | 20.6 | 117.1 | SASDB26 |
| MAG$_{1-5}$ N406Q glycosylated | 63.7 | 33.3 | 293 | 7.3 | 82.0 | 25.6 | 193.2 | SASDB36 |
| MAG$_{1-3}$ glycosylated | 39.8 | 43.7 | 293 | 3.9 | 43.0 | 13.0 | 59.3 | SASDB46 |
| MAG$_{1-5}$ wt deglycosylated | 56.8 | 37.5 | 293 | 7.3 | 75.4 | 25.5 | 166.2 | SASDBF6 |
| MAG$_{1-5}$ I473E deglycosylated | 56.8 | 45.1 | 293 | 6.0 | 61.1 | 21.2 | 99.6 | SASDB56 |
| MAG$_{1-5}$ N406Q deglycosylated | 56.8 | 33.6 | 277 | 7.8 | 93.9 | 29.0 | 216.4 | SASDB66 |
| MAG$_{1-3}$ deglycosylated | 36.6 | 38.2 | 293 | 3.9 | 39.9 | 12.6 | 49.4 | SASDB76 |

from aggregation during the experiment. Similar to the SAXS analysis, the MAG$_{1-5}$ I473E and MAG$_{1-3}$ SE-AUC data fit best to a single species that agrees with the $M_m$ of a monomer (Table 3). For wt MAG$_{1-5}$ and deglycosylated MAG$_{1-5}$, we could fit the data to a monomer–dimer equilibrium, with $K_d$s of $3.8 \times 10^2$ and $1.7 \times 10^2$ $\mu$M, respectively (Table 3). On the basis of a monomer–dimer equilibrium with these $K_d$s, dimer fractions of 18% for glycosylated wt MAG$_{1-5}$ and 24% for deglycosylated wt MAG$_{1-5}$ are expected to be present in the SAXS experiments (calculated at 52.2 and 37.5 $\mu$M for glycosylated and deglycosylated MAG, respectively). Indeed, the presence of a mix of monomers and dimers is observed in the SAXS data of both glycosylated and deglycosylated MAG$_{1-5}$ (Fig. 2j, Supplementary Fig. 11). The lack of dimers in the AUC experiments for MAG$_{1-5}$ I473E and MAG$_{1-3}$ and the higher affinity for deglycosylated MAG$_{1-5}$ compared with glycosylated MAG$_{1-5}$ further confirm that MAG forms dimers via domains Ig4 and Ig5 (Table 3).

**Structural basis of ligand recognition by Ig1.** MAG binds to sialic acids of gangliosides with its N-terminal V-type Ig domain and has a preference for a Neu5Ac-$\alpha$2,3-Gal-$\beta$1,3-GalNAc moiety[18]. We observed unmodelled electron density in the $2F_o - F_c$ and $F_o - F_c$ maps of the deglycosylated MAG$_{1-5}$ structure close to R118 in the putative ligand-binding site[20] (Fig. 3a). Native mass spectrometry of purified MAG$_{1-5}$ revealed a mixture of free and two ligand-bound MAG$_{1-5}$ forms with mass differences of $835 \pm 2$ and $854 \pm 2$ Da compared with free MAG$_{1-5}$ (Fig. 3b). The 835 Da ligand possibly corresponds to a tetrasaccharide comprising the aforementioned Neu5Ac-$\alpha$2,3-Gal-$\beta$1,3-GalNAc trisaccharide plus another hexose. The 854 Da ligand might correspond to a similar tetrasaccharide where Neu5Ac is replaced by Neu5Gc, a mammalian sialic acid variant not produced by humans. These ligands are likely co-purified in complex with MAG from the expression medium that contains beef digest (Primatone). We do not observe clear electron density for any ligands in the two other crystal forms (lysine-methylated MAG$_{1-5}$ and MAG$_{1-3}$). In these crystal forms the unliganded MAG is apparently preferentially crystallized over the ligand-bound forms. The resolution of the MAG$_{1-5}$–ligand complex (to 3.8 Å) is not sufficient to determine the detailed structure of the bound oligosaccharide. However, the unmodelled electron density is compatible with either of the suggested compounds.

To obtain more detailed information on MAG–ligand interactions, the commercially available trisaccharide 3′-N-Acetylneuraminyl-N-acetyllactosamine (Neu5Ac-$\alpha$2,3-Gal-$\beta$1, 4-GlcNAc) was soaked into the MAG$_{1-3}$ crystals, as these

provided higher resolution data. For the soaked crystals, diffraction data was collected to 2.3 Å resolution. The crystals were isomorphous to the unsoaked MAG$_{1-3}$ crystals (Table 1) and a $F_o$(soaked) $- F_o$(unsoaked) map revealed clear electron density in the oligosaccharide binding site for one of the two MAG$_{1-3}$ chains in the asymmetric unit (Fig. 3c). Most likely the other chain remained unliganded due to occlusion of the binding site by crystal packing.

The ganglioside binding side is formed by the CC′ loop and the F and G $\beta$-strands of the N-terminal V-type Ig1 domain. In agreement with previous data[20], the side chain of R118 in strand F forms a key salt bridge with the carboxylic acid group of the sialic acid in the trisaccharide ligand. Y65 in the CC′ loop forms extensive Van der Waals' contacts with the ligand as well as a hydrogen bond to the O9 of the Neu5Ac sialic acid. Other contributing interactions are made by: (1) the backbone carbonyls of N126 and T128 in strand G that form hydrogen bonds with H-N5 and the H-O9 of the sialic acid respectively, (2) the backbone amide proton of T128 that forms a hydrogen bond with the O8 of the sialic acid and (3) the side chain of Y127 that forms Van der Waals' contacts with the glycerol group (C7–C9) of the sialic acid (Fig. 3d).

We validated the ganglioside binding properties of MAG by mutating ligand-binding residues to alanine in MAG$_{1-5}$ and probed their interaction with GT1b ganglioside incorporated into liposomes. The set-up we used, MAG$_{1-5}$ coupled at the C-terminus to a streptavidin-coated surface plasmon resonance (SPR) chip and GT1b-containing liposomes in the mobile phase, enables avidity-enhanced interactions that also occur in *trans* between cells (Fig. 3f, see 'Methods' section). Indeed we observed specific binding of GT1b liposomes to wt MAG$_{1-5}$, no interactions with the ligand-binding mutants MAG$_{1-5}$ R118A, T128A and Y127A and reduced interaction with MAG$_{1-5}$ Y65A (Fig. 3f). In addition, the MAG$_{1-5}$ W25Q mutant that lacks the tryptophan mannosylation on W22, still interacted with GT1b liposomes in this assay. Remarkably, this W25Q mutant appeared to have higher affinity for the GT1b liposomes compared with wt MAG. This suggests that rather than contributing to the interaction strength, this tryptophan mannosylation on W22 may play a regulatory role in binding (membrane-embedded) gangliosides. In summary, we have shown that MAG interacts with membrane-bound gangliosides via the side chain of R118, the CC′-loop and the F and G $\beta$-strands of the N-terminal V-type Ig1 domain and that the W22 mannosylation does not enhance ganglioside binding.

Ligand interaction of MAG is similar to the sialic acid recognition of other Siglec family members (Fig. 3g)[30,32]. As in

**Table 3 | SE-AUC parameters.**

| Sample | Model: | $M_m$ floated/fixed | $M_m$ (kDa) | $\log_{10} (K_a)$ | $K_d$ (µM) | $\chi^2$ |
|---|---|---|---|---|---|---|
| MAG$_{1-5}$ wt glycosylated | Monomer–dimer equilibrium | Floated | 62.6 | 3.42 | 382 | 1.17 |
| MAG$_{1-5}$ I473E glycosylated | Single species | Floated | 62.3 | – | – | 1.18 |
| MAG$_{1-3}$ glycosylated | Single species | Floated | 40.8 | – | – | 1.31 |
| MAG$_{1-5}$ wt deglycosylated | Monomer–dimer equilibrium | Fixed | 56.9 | 3.78 | 167 | 0.94 |

Siglec5 (ref. 32), the MAG Ig1 CC′-loop (residues 64–70) seems to undergo conformational selection on ligand binding. This loop adopts a single conformation when ligand is bound, whereas it can have several conformations (including the ligand-bound conformation) or is unstructured in the different unliganded MAG crystal forms (Fig. 3e). Furthermore, this loop adopts different conformations in the Siglec -1, -5 and -7 structures (both unliganded and ligand-bound forms, Fig. 3g)[30–32]. The combination of our structural and biophysical data on MAG–ganglioside interaction, with that of others on MAG's specificity for Neu5Ac-α2,3-Gal-β1,3-GalNAc (ref. 18) establishes the structural basis of ganglioside recognition by MAG.

**Neurite outgrowth inhibition depends on MAG dimerization.** We tested different MAG variants in neurite outgrowth assays to determine the role of MAG dimerization for neuronal plasticity inhibition (Fig. 4). In agreement with previous data[20,38], MAG$_{1-5}$ wt on coverslips inhibited neurite outgrowth of hippocampal neurons compared with poly-L-lysine (PLL)-covered slips (Fig. 4a,b). Other dimeric variants (MAG$_{1-5}$ N406Q and MAG$_{1-5}$-Fc) inhibited neurite outgrowth to a similar level (Fig. 4d,f). The monomeric MAG$_{1-5}$ I473E and MAG$_{1-3}$ wt on the other hand showed no significant neurite outgrowth inhibition (Fig. 4c,e). Interestingly, MAG$_{1-5}$-Fc R118A, which is dimeric but lacks sialic acid binding properties, showed neurite outgrowth stimulation instead of inhibition, compared with PLL (Fig. 4g). These data indicate that dimerization through domains Ig4–Ig5 and the ability to bind sialic acid moieties on the neuronal surface are required for neurite outgrowth inhibition signalling by MAG for hippocampal neurons.

**Discussion**
MAG controls adhesion and signalling between myelinating cells and axons. In contrast to earlier studies[28,29], we find that MAG does not fold back onto itself like an Ig-horseshoe as in the L1CAM and axonin neuronal adhesion molecules. Instead, our data show that the extracellular region of MAG has an extended shape with limited inter-domain flexibility, similar to several other cell adhesion molecules such as SYG, Cadherin and Nectin family members[39–41].

The structure of MAG is the first of a full extracellular portion of a Siglec family member. Besides the common N-terminal V-type Ig domain for recognizing sialic acid moieties, Siglecs vary in the number of additional Ig domains; from 1 up to 16. Comparison of the structures of MAG and Siglec5 reveals a different inter-domain orientation between domains Ig1 and Ig2, likely due to differences in amino acids at the interface (Supplementary Fig. 17 and Supplementary notes).

By binding to axonal gangliosides, MAG maintains a defined spacing between the innermost myelin surface and the axon surface[10–12,42]. This myelin–axon spacing has been reported to be 12–14 nm based on electron micrographs of chemically fixed myelin tissue[10–12]. However, analysis of more recent electron micrographs of high-pressure frozen myelin that does not suffer from fixation induced artefacts[43] reveals an axon–myelin spacing

of 9–12 nm. This periaxonal diameter matches well with a straightforward model that follows from our structural data of the MAG dimer and the MAG–ganglioside interaction; two opposing membrane surfaces are spaced 10 nm apart when the membrane-proximal C-termini of the MAG dimer are positioned on one membrane (the structures lack only two residues to the transmembrane helix) and the MAG dimer-bound gangliosides are positioned on the other membrane (Fig. 5). Although the two crystal forms of the full extracellular segment of MAG have different inter-domain angles (3.4–17.4°), the overall arrangement and resulting structure-based axon–myelin spacing is similar (see Fig. 2a). The agreement of intermembrane distance determined from high-pressure frozen EM on myelin tissue[43] and here by structural and biophysical data on the extracellular segment of MAG indicates that in the periaxonal space, MAG is dimerized *in cis* via domains Ig4–Ig5 when bound to axonal gangliosides *in trans*.

Intriguingly, in this model, the unusual tryptophan C-mannosylation on W22 is positioned at the interface of MAG and the extracellular leaflet of the axonal membrane (Fig. 5, Supplementary Fig. 18). The WxxW motif is conserved among MAG orthologues in vertebrates from fish to human, but is absent in all other Siglec paralogs (Supplementary Fig. 6). This suggests that tryptophan mannosylation is specific for the function of MAG. We showed that the mannosyl group does not enhance the binding of MAG to GT1b ganglioside liposomes but may weaken it. Possibly, tryptophan mannosylation of MAG provides specificity to sialylated ligand interactions. Alternatively, the close proximity of the mannosyl group to the axonal membrane during MAG–ganglioside interaction may indicate a regulatory role in axonal membrane engagement.

The buried-surface area of the MAG dimerization interface formed by Ig4 and Ig5 is large (2,037 Å$^2$) and hydrophobic. We find, however, that the affinity of MAG dimerization via this interface in solution is low ($K_d$ of $3.8 \times 10^2$ µM). The weak interaction is probably important in the native context, where MAG is expressed on the cell surface, as the *cis* dimer can be *trans* stabilized by interaction with gangliosides on the opposing axonal membrane. For the N-Cadherin family of cell adhesion molecules it has been shown that affinities as weak as 10 mM in solution are functionally important in the context of a *trans*-stabilized *cis*-dimer[39].

Dimerization of MAG may serve two purposes. It provides a mechanism to restrain the intermembrane distance, since a *cis–trans* stabilized MAG dimer would restrict angular freedom with respect to the membrane more than a *trans* only stabilized MAG monomer. In this sense, the MAG dimer could function as a 'molecular leaf spring' that maintains the well-defined spacing between the axonal membrane and the adaxonal myelin membrane along the internode. In addition, dimerization of MAG could enable compaction of the periaxonal space. The weak *cis*-interaction of MAG, if not stabilized in *trans*, may ensure enough monomer is available to bridge a wider periaxonal spacing (of up to 16 nm, based on the length of a MAG monomer) that may exist during myelin formation. Even greater distances could be bridged if MAG binds to sialylated N- or O-linked glycans of axonal surface glycoproteins before reaching

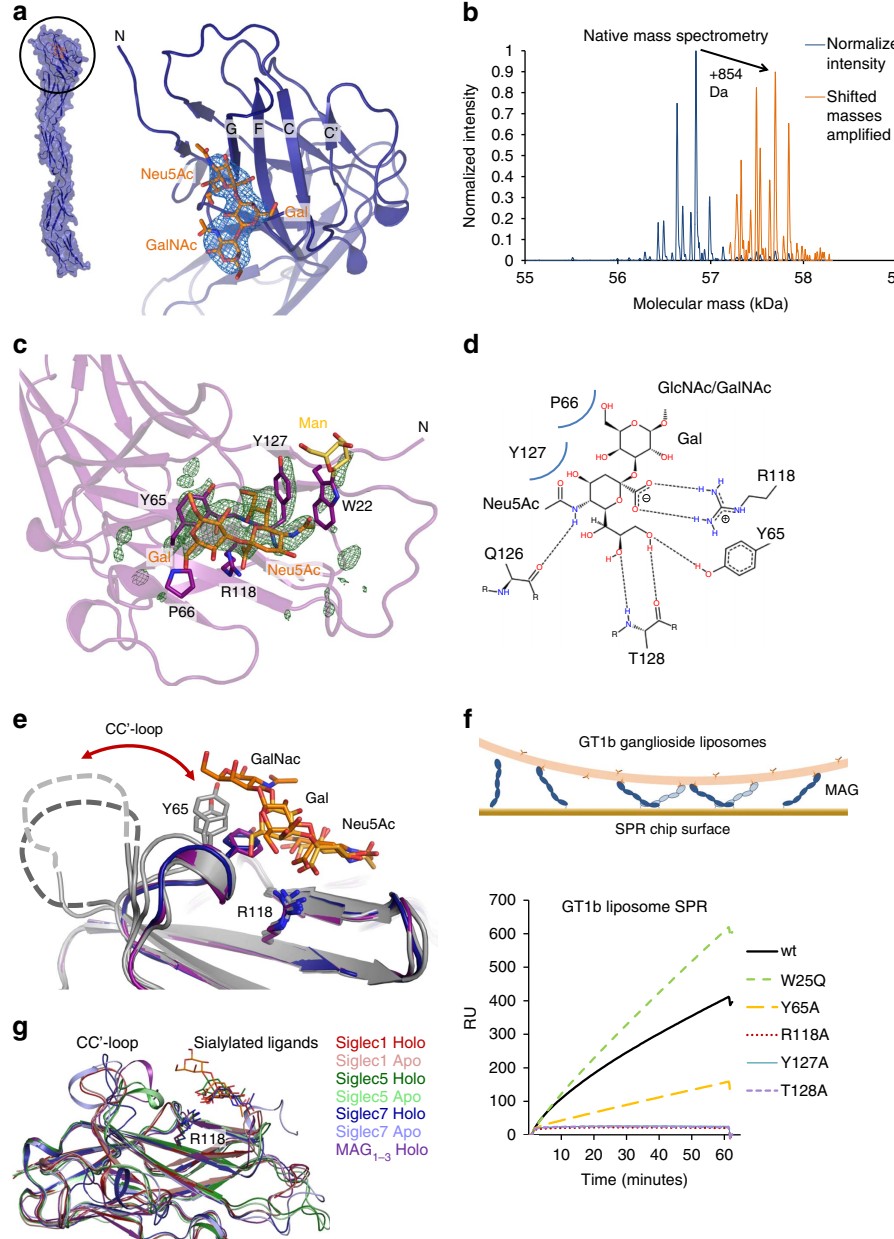

**Figure 3 | Structural characterization of ligand recognition by the N-terminal domain Ig1 of MAG.** (**a**) $2F_o - F_c$ electron density map displayed at a contour level of 1.3 $\sigma$ before placing any ligand for refinement in the $MAG_{1-5}$ deglycosylated crystal structure (blue), showing a density that fits well with the Neu5Ac-α2,3-Gal-β1,3-GalNAc trisaccharide (orange). (**b**) Native mass spectrometry reveals two species that have a mass difference of 854 ± 1.4 Da, presumably because of oligosaccharide ligand binding. The deconvolved mass versus intensity spectrum (blue) is shown together with the 20 × amplified version of this spectrum for masses above 57.2 kDa (orange) to highlight the similar pattern of trimmings between the unliganded and ligand-bound form. (**c**) $F_o - F_o$ (soaked–unsoaked) electron density at a contour level of 3.0 $\sigma$ of $MAG_{1-3}$ at the ganglioside binding site of chain B, showing the unbiased electron density changes that resulted from binding of the 3'-Sialyl-N-acetyllactosamine (Neu5Ac-α2,3-Gal-β1,4-GlcNAc) ligand and concomitant small conformational rearrangements. Residues involved in ligand engagement (sticks), as well as the C-mannosylation (yellow) on W22 are shown. The first two sugars of the 3'-Sialyl-N-acetyllactosamine (orange) fit the density well. (**d**) Protein-ligand interactions with hydrogen bonds indicated by dashes and Van der Waals' contacts by curved blue lines. (**e**) Comparison of the four unliganded (grey) and the two ligand-bound structures of MAG; $MAG_{1-5}$ deglycosylated (blue) and soaked $MAG_{1-3}$ (purple). The CC' loop adopts different conformations in the unliganded structures yet appears to undergo conformational selection by interactions of Y65 in this loop with the ligand. (**f**) GT1b ganglioside liposome SPR confirms the contribution of contact residues from the crystal structures. Liposome and MAG molecules are displayed approximately to scale in schematic representation. Surprisingly, the W25Q mutant that lacks the tryptophan mannosylation on W22 shows enhanced ligand binding. (**g**) Similar ligand recognition by four different Siglec family members; MAG (purple, $MAG_{1-3}$ structures), Siglec1 (red), Siglec5 (green) and Siglec7 (blue). Shown are unliganded forms (lighter colours), ligand-bound forms (darker colours) with sialylated ligands and the conserved arginine (stick representation, R118 in MAG) that forms a salt bridge with the carboxylic acid group of the sialic acid. The structurally heterogeneous CC' loop is also indicated.

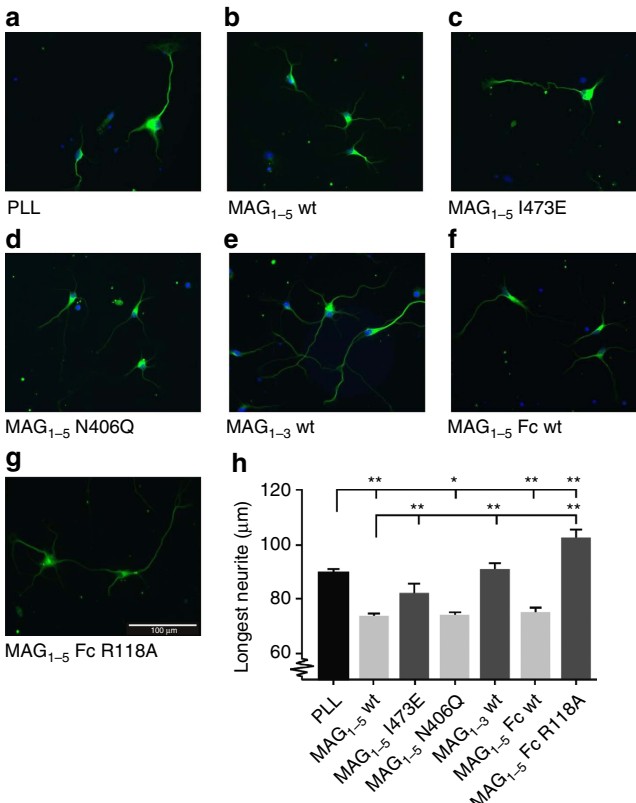

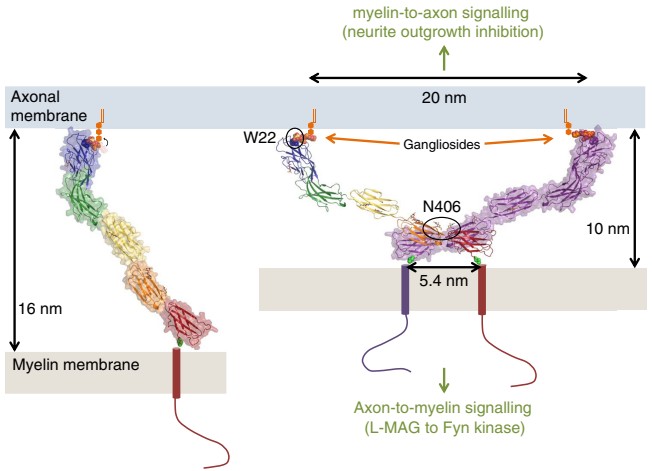

**Figure 5 | Model for MAG-mediated myelin–axon engagement and signalling.** The MAG$_{1-5}$ (cartoon and surface representation) monomer and dimer are indicated, with the trisaccharide ligand (orange spheres, colouring as in Fig. 1) and the tryptophan mannosylation on W22 (yellow spheres). Whereas monomeric MAG could span intermembrane distances of ~16 nm (left), the *cis* dimerization of MAG restricts the periaxonal diameter to 10 nm (right). The arrow at the ganglioside binding site in the left panel indicates that the third sugar, which has the highest B-factors in the deglycosylated MAG$_{1-5}$ structure and is not visible in the ligand-bound MAG$_{1-3}$ structure, needs to be in a different conformation to accommodate such binding. The dimerization of MAG brings the cytosolic regions into close proximity. This possibly triggers signalling into the myelin-forming oligodendrocyte cell. See also Supplementary Fig. 18.

**Figure 4 | MAG dimerization and sialic acid-binding are required for neurite outgrowth inhibition of hippocampal neurons.** Dissociated neuron cultures were prepared from P1 hippocampus and grown for 2 days *in vitro* (DIV2) on (**a**) PLL (20 µg ml$^{-1}$) as a control or on coverslips coated with PLL and MAG proteins (60 µg ml$^{-1}$): (**b**) MAG$_{1-5}$ wt, (**c**) MAG$_{1-5}$ I473E, (**d**) MAG$_{1-5}$ N406Q, (**e**) MAG$_{1-3}$ wt, (**f**) MAG$_{1-5}$-Fc wt or (**g**) MAG$_{1-5}$-Fc R118A. Cultures are immunostained with antibodies against βIII-tubulin and DAPI. Quantification of the length of the longest neurite is shown in µm in (**h**). $n = 6$ mice, One-way ANOVA with Tukey *post hoc* test was used, $^*P < 0.05$. $^{**}P < 0.01$. Error bars represent s.e.m. Only dimeric and not monomeric MAG constructs inhibit neurite outgrowth compared with PLL. Surprisingly, a dimeric variant that does not bind sialic acids (MAG$_{1-5}$-Fc R118A) appears to stimulate neurite outgrowth.

the gangliosides. When *trans* interactions with gangliosides have been established and possibly local concentrations are elevated due to the abundance of ganglioside ligands, formation of MAG *cis*-dimers is triggered concomitant with compaction of the periaxonal diameter to its final spacing (Fig. 5).

Dimerization of MAG can be regulated by modulating glycosylation on N406. Large and charged glycans on N406 can obstruct dimerization because of steric clashes and coulombic repulsion. Trimming all MAG N-linked glycans down to single N-Acetylglucosamines by deglycosylation with Endo-H$_f$ or preventing glycosylation on N406 by mutating it to glutamine enhances dimerization (Fig. 2, Supplementary Figs 8–12 and 16, Tables 2 and 3). This glycosylation site might play a regulatory role during myelin development and myelination-related pathologies. MAG glycosylation changes during development[44] and abnormal glycosylation of MAG correlates with myelination deficiencies[45–47]. Possibly, modulation of N406 glycosylation, either at the biosynthesis level or by extracellular trimming, affects MAG dimerization and thereby impacts on the myelin–axon interaction (see Supplementary notes for details).

Myelin-to-neuron signalling with MAG as a ligand can inhibit neurite outgrowth[2,3,20,21,48]. Studies based on MAG truncations,

chimeras and mutant versions suggested that the inhibitory properties reside in domains Ig4 and Ig5, and in the sialic acid binding site[20,49,50]. For example, a chimeric protein consisting of Siglec1 Ig domains 1–3 plus MAG Ig4-5, but not Siglec1 domains 1–3 alone, has neurite outgrowth inhibition properties similar to MAG[49]. We confirmed that the sialic acid-binding properties of MAG are required for neurite outgrowth inhibition in hippocampal neurons as the MAG$_{1-5}$-Fc R118A mutant that lacks ganglioside binding properties does not inhibit, but surprisingly, stimulates neurite outgrowth (Fig. 4g). This has not been reported before and may be an interesting new avenue for therapeutic intervention to enhance central nervous system regeneration. In addition we show that domains Ig4 and Ig5 are essential for MAG dimerization and that neurite outgrowth inhibition is abrogated by the I473E point mutation that monomerizes MAG (Fig. 4e,c). Thus, it is the dimerization of MAG that is required for neurite outgrowth inhibition, rather than direct interactions of domains Ig4 and Ig5 with neuronal receptors, as previously suggested[49,50]. Although other protein receptors have been identified that mediate the neurite outgrowth-inhibiting signalling by MAG[3], it has been shown that direct clustering of gangliosides by antibodies also leads to inhibition of neurite outgrowth of hippocampal neurons[20]. The combination of our data and that of others indicates that MAG dimerization at domains Ig4 and Ig5 and sialic acid binding at domain Ig1 induces neurite outgrowth inhibition for hippocampal neurons, by clustering of gangliosides.

Axon-to-myelin signalling with MAG as a receptor controls myelin formation. Antibody-mediated extracellular clustering of the L-MAG isoform activates Fyn kinase[23] and Fyn activation is essential for the initiation of myelination[25]. Our structures show that the C-termini of MAG$_{1-5}$ are separated by 5.4 nm in the dimer, bringing the cytosolic regions into close proximity (Fig. 5). Probably, L-MAG dimerization as a result of *trans* interaction

with gangliosides on the axon brings the cytosolic regions of MAG into close proximity to trigger activation of Fyn, similar to Fyn activation by signalling lymphocytic activation molecule clustering in immune cells[51]. Whether MAG forms higher-order clusters that are triggered by dimerization needs to be established, but preference of both MAG and Fyn for lipid rafts[24,52] suggests that both proteins can be locally enriched in the membrane to assist clustering.

## Methods

**Generation of constructs and mutagenesis.** MAG constructs were generated by polymerase chain reaction (PCR) using mouse S-MAG (IMAGE 40039200) as a template and primers to start at (UNIPROT) residue number 20 (after the signal peptide) and end at residue 325 for $MAG_{1-3}$ and residue 508 for $MAG_{1-5}$. Point mutants were also created by PCR, either by a two-step PCR with overlapping primers (W22A, W25Q, Y65A, R118A, Y127A, T128A, I473E,) or by a single-step PCR using non-overlapping phosphorylated primers (N406Q). All constructs were subcloned using BamHI/NotI sites in pUPE107.03 (cystatin secretion signal peptide, C-terminal His$_6$-tag), unless indicated otherwise.

**Large-scale expression and purification.** Constructs were transiently expressed in N-acetylglucoaminyltransferase I-deficient (GnTI − ) Epstein–Barr virus nuclear antigen I (EBNA1)-expressing HEK293 cells in suspension (U-protein express). Medium was collected 6 days after transfection and cells were spun down by 10 min of centrifugation at 1,000 g. Supernatant was concentrated fivefold and diafiltered against 500 mM NaCl, 25 mM HEPES pH 7.8 (IMAC A) using a Quixstand benchtop system (GE Healthcare) with a 10 kDa molecular weight cut-off (MWCO) membrane. Cellular debris was spun down for 10 min at 9,500 g and the concentrate was filtered with a glass fibre prefilter (Minisart, Sartorius). Protein was purified by Nickel–nitrilotriacetic acid (Ni–NTA) affinity chromatography followed by SEC on a Superdex200 Hiload 16/60 column (GE Healthcare) equilibrated in SEC buffer (150 mM NaCl, 20 mM HEPES pH 7.5). Protein was concentrated to 7–14 mg ml$^{-1}$ using a 30 kDa MWCO concentrator before plunge freezing in liquid nitrogen and storage at − 80 °C.

**Crystallization and data collection.** Since initial crystallization attempts did not yield diffraction-quality crystals, several methods were used to enhance crystallization. Deglycosylation was performed by adding Endo-H$_f$ (1.0 × 10$^6$ U ml$^{-1}$, New England Biolabs) 1:100 (v per v) directly to the concentrated protein and incubating overnight at 37 °C. Completeness of the reaction was analysed by SDS-PAGE and sample quality was assessed by SAXS (see Table 2). As an alternative approach, reductive lysine methylation was performed on glycosylated $MAG_{1-5}$ diluted to 1 mg ml$^{-1}$, by two steps of 2 h incubation at 4 °C with 1 M dimethylamine–borane complex (added 1:50, v per v), and 1 M formaldehyde (added 1:25, v per v)[53]. The reaction was completed by a final addition of 1 M dimethylamine–borane complex (added 1:100, v per v) and incubated overnight at 4 °C, after which the reaction was quenched by performing a SEC run on a Superdex200 (GE Healthcare) column equilibrated in 20 mM Tris/HCl pH 7.5, 200 mM NaCl. Therefore, this was the buffer used to set-up crystallization experiments of the lysine-methylated protein. Sitting-drop vapour diffusion at 18 °C was used for all crystallization trials, by mixing 150 nl of protein solution with 150 nl of reservoir solution. Crystals of deglycosylated $MAG_{1-5}$ (6.7 mg ml$^{-1}$) appeared in a condition of 100 mM NaCl, 20 mM Tris/HCl pH 7.0, 7.7% PEG 4,000 (w per v). Crystals of lysine-methylated $MAG_{1-5}$ (8.4 mg ml$^{-1}$) appeared in a condition containing 200 mM NaOAc, 20% PEG3350 (w per v). Crystals of $MAG_{1-3}$ (12.2 mg ml$^{-1}$) appeared in a condition containing 0.05 M tri-sodium citrate, 1.2 M ammonium sulfate, 3% (w per v) isopropanol. $MAG_{1-3}$ crystals were soaked by addition of 1 µl reservoir solution containing 10 mM 3′-N-acetylneuraminyl-N-acetyllactosamine (Neu5Ac-α2,3-Gal-β1,4-GlcNAc, Sigma-Aldrich product A6936) to the drop. Crystals were cryo-protected with reservoir solution supplemented with 25% of glycerol for deglycosylated $MAG_{1-5}$ and $MAG_{1-3}$ and with 25% ethylene glycol for lysine-methylated $MAG_{1-5}$. After brief incubation in the cryo-protectant solution, crystals were plunge-cooled in liquid nitrogen. Data sets were collected at 100 K at the Deutsches Elektronen-Synchrotron PETRA III beamline P14 (lysine-methylated $MAG_{1-5}$, $\lambda = 0.97553$ Å), the European Synchrotron Radiation Facility (ESRF) beamline ID23-1 ($MAG_{1-3}$, $\lambda = 0.97599$ Å) and Swiss Light Source beamline PX (deglycosylated $MAG_{1-5}$ and soaked $MAG_{1-3}$, $\lambda = 0.99998$ Å).

**Structure solution and refinement.** Data were integrated by IMOSFLM[54] ($MAG_{1-5}$ deglycosylated, $MAG_{1-5}$ lysine-methylated and $MAG_{1-3}$ ligand bound) or XDS[55] ($MAG_{1-3}$ unliganded) and scaled and merged by the AIMLESS pipeline[56]. All structures were solved by molecular replacement using PHASER[57]. Initial search models were PDB IDs 1URL (ref. 58) for Ig1, 4FRW (ref. 41) residues 150–242 for Ig2, 1CS6 (ref. 59) residues 308–388 for domain Ig3, 3P3Y (ref. 60) residues 55-185 for Ig4 and 2YD6 (ref. 61) residues 132–221 for Ig5. First $MAG_{1-3}$ was solved by searching for two copies of 1URL, followed by two copies of 1CS6

and finally two copies of 4FRW. Search models were trimmed to polyalanine chains by CHAINSAW (ref. 62) to obtain better starting density. Next, deglycosylated $MAG_{1-5}$ was solved by searching for the first two (refined) domains of $MAG_{1-3}$, followed by SCULPTOR (ref. 63)-trimmed versions of 3P3Y and 2YD6 respectively. Lastly, the Ig3 domain was searched using the refined Ig3 of MAG from $MAG_{1-3}$ as this domain had a much higher B-factor in the $MAG_{1-5}$ crystals, likely because of the lack of any crystal packing contacts for this domain. It was important to realize that the unit cell only contained a single copy of MAG, resulting in a solvent content of 91%, which was used in PHASER to estimate the total scattering. Lysine-methylated $MAG_{1-5}$ was solved by searching for the Ig1, Ig2, Ig4, Ig5 and Ig3 from the refined $MAG_{1-3}$ and deglycosylated $MAG_{1-5}$ structures, in that order. Again, Ig3 was searched last because of the high B-factor. After molecular replacement, models were improved by iterative density modification by DM (CCP4)[64], manual model building in COOT[65] and refinement with REFMAC[66]. Final refinement was performed with PHENIX[67] and validation with Molprobity[68]. In both $MAG_{1-5}$ structures, Ig3 from the higher resolution $MAG_{1-3}$ was used as a reference structure to restrain refinement, because of the high B-factor and the resulting poor density for that domain. Ramachandran statistics were (% Ramachandran favored/% allowed/% outliers): 93/7/0 ($MAG_{1-5}$ deglycosylated), 91/9/0 ($MAG_{1-5}$ lysine-methylated), 97/3/0 ($MAG_{1-3}$ unliganded) and 96/4/0 ($MAG_{1-3}$ ligand bound). The buried surface are for MAG dimerization was calculated by PISA[69].

**Small angle X-ray scattering.** SAXS was performed at the ESRF BM29 BioSAXS beamline equipped with a 2D Pilatus 1 M detector (DECTRIS, Switzerland), operated at an energy of 12.5 keV. MAG constructs were diluted with and dialyzed against SEC buffer using a 10 kDa MWCO membrane. The concentrations were determined by ultraviolet–visible spectroscopy on a nanodrop ND-1,000 spectrophotometer. Similar concentrations were selected for all samples to allow comparison (see Table 2). SAXS data were collected at 20 °C unless indicated otherwise. The data were radially averaged, normalized to the intensity of the transmitted beam and exposure time and the scattering of the solvent-blank (SEC buffer) was subtracted, following standard procedures. The curve was scaled using a BSA reference so that the $I_0$ represents the $M_m$ of the sample. Radiation damage was monitored by comparing curves collected from the same sample, only curves without radiation damage were merged. A single concentration was used for all measurements, no extrapolation to zero concentration was performed. Data were analysed by PRIMUS[70], GNOM[71], DAMMIF[72], CRYSOL[73] and OLIGOMER[70] of the ATSAS[74] suite.

**Analytical ultracentrifugation.** Sedimentation equilibrium experiments were carried out in a Beckman Coulter Proteomelab XL-I and a Beckman Optima XL-A analytical ultracentrifuge. Either 12 or 3 mm centerpieces with quartz windows were used, 12 mm for the lowest concentrations and 3 mm for the others. An-60 and An-50 Ti rotors (Beckman) were used to carry out the measurements. MAG constructs were diluted with and dialyzed against SEC buffer using a 10 kDa MWCO membrane. Input concentrations of 3.8, 35.3 and 89.0 µM ($MAG_{1-5}$ wt glycosylated), 16.8, 27.2, 35.5 and 144 µM ($MAG_{1-5}$ wt deglycosylated), 3.6, 8.3, 41.4 and 102.8 µM ($MAG_{1-5}$ I473E glycosylated), 7.2, 57.4 and 332 µM ($MAG_{1-3}$ wt glycosylated) were used. Sedimentation equilibrium runs were performed at 20 °C and at 7,500, 14,000 and 20,000 r.p.m. Absorbance was determined at 250, 280 and 300 nm with SEC buffer as reference. Buffer density and viscosity were determined by SEDNTERP as 0.99823 g ml$^{-1}$ and 0.001002 Pa s, respectively.

**In-gel digestion and LC-MS/MS.** $MAG_{1-5}$ was separated by SDS-PAGE, and cut from gel for digestion with trypsin (Promega). The gel band was cut to small pieces, washed in Milli-Q water and treated with acetonitrile to shrink the gel pieces. The sample was then incubated in 1 g l$^{-1}$ 1,4-dithiothreitol for 60 min at 60 °C, treated with acetonitrile, alkylated with 10 g l$^{-1}$ iodoacetamide for 30 min at room temperature in the dark and subsequently washed with ammonium bicarbonate and treated with acetonitrile, twice. The gel pieces were then incubated on ice for 90 min with 30 mg l$^{-1}$ trypsin. Excess trypsin was removed, the gel pieces were covered in ammonium bicarbonate, and the samples were subsequently incubated overnight at 37 °C. The digested samples were collected and remaining sample was extracted from the gel pieces by treatment with acetonitrile. The solution with the peptides was subsequently dried in a speedvac and the peptides resuspended in 10% formic acid, 5% dimethylsulfoxide in water.

Peptides were separated by reversed phase LC coupled on-line to an Orbitrap Elite for MS/MS analysis. The nano-LC consists of an Agilent 1200 series LC system equipped with a 20 mm ReproSil- Pur C18-AQ (Dr Maisch GmbH) trapping column (packed in-house, i.d., 100 µm; resin, 5 µm) and a 400 mm ReproSil-Pur C18-AQ (Dr Maisch GmbH) analytical column (packed in-house, i.d., 50 µm; resin, 3 µm) arranged in a vented-column configuration. The flow was passively split to 100 nl min$^{-1}$. We used a standard 45 min gradient from 7–30% acetonitrile in aqueous 0.1% formic acid. All precursors were fragmented by both ETcaD and HCD. Data were searched against a custom database of recombinant protein sequences, including the MAG constructs used here, with trypsin as protease, allowing up to two missed cleavages. We used a 50 p.p.m. precursor mass window and 0.02 Da fragment mass window. The C-mannosylated peptide, with

3 + precursor charge, eluted after 33–34 min and was identified by both HCD and ETcaD MS/MS, with matched fragment ions supporting site localization for the C-mannosylation.

**Native mass spectrometry.** Purified protein samples were deglycosylated with Endo-H$_f$ as for crystallization, followed by buffer exchange to 150 mM ammonium acetate pH 7.5, using Vivaspin500 10 kDa MWCO centrifugal filter units. Samples were loaded onto gold-coated borosilicate capillaries prepared in-house for nanoelectrospray ionization. Samples were analysed on a modified Orbitrap extended mass range (Thermo Fisher) for high mass ions[75].

**Surface plasmon resonance.** MAG$_{1-5}$ wt and mutants cloned in-frame with an N-terminal cystatin secretion signal and a C-terminal biotin acceptor peptide and His$_6$-tag (sequence AAAGSGLNDIFEAQKIEWHEGRTKHHHHHH), were biotinylated in HEK293 cells by co-transfection with E. coli BirA biotin ligase with a sub-optimal secretion signal (in a pUPE5.02 vector), using a DNA ratio of 9:1 (MAG DNA: BirA DNA, m per m)). Additional sterile biotin (100 μl of 10 mg ml$^{-1}$ Tris-buffered biotin per 4 ml HEK293 culture) was supplemented to the medium a few hours after transfection. MAG mutants were purified from the medium by Ni–NTA affinity purification. Purity was evaluated by SDS-PAGE and coomassie staining and biotinylation by a streptavidin gel-shift assay followed by α-His$_6$ Western blot (Supplementary Fig. 19). C-terminally biotinylated MAG proteins were spotted on a G-STREP SensEye chip (Ssens) with a Continuous Flow Microspotter (Wasatch Microfluidics) using a 8 × 6 format. The C-terminal coupling of MAG to the surface mimics the native, membrane attached topology. SEC buffer with 0.005% tween was used as a spotting buffer and the coupling was quenched using 1 mM biotin in SEC buffer solution.

GT1b ganglioside liposomes were prepared as described previously[76]. In brief, the lipids dipalmitoyl phosphatidylcholine, dipalmitoyl phosphatidylglycerol, cholesterol and GT1b gangliosides were mixed in a molar ratio of 40.3:4.2:40.9:1.3 in a chloroform/methanol mixture (6:1, v per v). The lipid mixture was dried under vacuum on a rotary evaporator to create a thin film of lipids. Liposomes were formed by addition of 1 ml of SEC buffer per 21.7 μmol of lipid mixture and 11 freeze-thaw cycles. As a negative control, liposomes with the same lipid composition but lacking GT1b were prepared using the same protocol. Liposomes were extruded through a 100 nm membrane with a mini-extruder (Avanti Polar Lipids) at 70 °C.

SPR experiments, with the liposomes in the mobile phase and the MAG constructs attached to the surface, were performed on an MX96 SPRi instrument (IBIS Technologies) equipped with a CX flowcell, using the CX vesicle run protocol and an association time of 60 min at a temperature of 25 °C. As a running buffer, SEC buffer without any detergent was used. Preliminary removal of co-purified ligands before the runs and regeneration after runs was performed by washes with 0.5% SDS in phosphate-buffered saline (PBS) followed by 5M NaCl. Data was zeroed and referenced using SprintX 1.11 (IBIS Technologies).

**Animals.** All animal use and care was in accordance with institutional guidelines and approved by the animal experimentation committee (DEC). Littermate C57BL/6 (Charles River) mice were killed by decapitation at postnatal day 1 (P1) before the brain was removed to prepare hippocampal neuronal cultures.

**Neurite outgrowth assays.** MAG$_{1-5}$-Fc constructs were generated by subcloning into pUPE7.12 vector using BamHI/NotI restriction sites (Fc is C-terminal of MAG), expressed in HEK293 cells and purified by protein-A affinity purification using standard protocols. Coverslips were all coated overnight at 4 °C with PLL(20 μg ml$^{-1}$) and 2 h at 37 °C with different MAG proteins (60 μg ml$^{-1}$): PBS (PLL control), MAG$_{1-5}$ wt, MAG$_{1-5}$ I473E, MAG$_{1-5}$ N406Q, MAG$_{1-3}$ wt, MAG$_{1-5}$-Fc wt or MAG$_{1-5}$-Fc R118A (at least three independent experiments were performed). It is expected that proteins immobilize non-specifically to the coverslips, therefore the immobilization efficiency of the different MAG variants was not experimentally verified. Hippocampal cultures were prepared as described previously[77]. In brief, the hippocampus was dissected at P1 and collected in L15 dissection medium (Gibco) containing 7 mM HEPES (L15-HEPES). Cells were dissociated by incubation in 0.25% trypsin in L15-HEPES at 37 °C for 20 min. Following three washes with L15-HEPES, cells were triturated using a fire-polished Pasteur pipette in growth medium (neurobasal medium (Gibco) with B-27 supplement (Thermo-Fisher), L-glutamine, penicillin/streptomycin and β-mercaptoethanol). Hippocampal neurons were resuspended in fresh growth medium and plated onto PLL- and MAG-coated glass coverslips. After two days in vitro (DIV2), hippocampal cultures were fixed for 10 min in 4% paraformaldehyde followed by three washing steps with PBS, blocking with 5% normal donkey serum and 0.1% triton in PBS, and incubated overnight at 4 °C with mouse anti-βIII-tubulin primary antibody (Covance, 1:500 (v per v)). The next day, cultures were washed three times with PBS and incubated with the secondary antibody (donkey anti-mouse Alexa Fluor 488, 1:750) for 2 h at room temperature. After three PBS washing steps, coverslips were incubated with 4′,6-diamidino-2-phenylindole (DAPI—Sigma) for 10 min. After several PBS washes, coverslips were mounted with FluorSave. Hippocampal neurons were visualized using a Zeiss Axioskop A1 using a 20 × objective. Images were analysed using FIJI (version

2.0.0)[78] by tracing the longest neurite of a hippocampal neuron (positive for tubulin and DAPI). Significance was determined using a one-way ANOVA ($P = 0.0001$) with a Bonferroni posthoc test for multiple comparisons (Graphpad Prism version 6.07). Every construct was compared with control PLL or MAG wt. All quantitative assessments in this manuscript were performed while being unaware of the condition to avoid observer bias. During analysis, raw data were named in a descriptive way, without revealing experimental group information. Fluorescent microscopic imaging was always done with the same settings within experiments, and analysing techniques were standardized. A single person analysed all data obtained within the experiments. Significant levels were reached for the following proteins: MAG$_{1-5}$ wt (**, 348 neurons counted), MAG$_{1-5}$ N406Q (*, 385 neurons counted), MAG$_{1-5}$-Fc wt (**, 364 neurons counted) and MAG$_{1-5}$-Fc R118A (*, 369 neurons counted) compared with PLL (395 neurons counted). Constructs MAG$_{1-5}$ I473E (*, 342 neurons counted), MAG$_{1-3}$ wt (***, 353 neurons counted) and MAG$_{1-5}$-Fc R118A (****, 348 neurons counted) were significant compared with MAG$_{1-5}$ wt. *$P < 0.05$, **$P < 0.01$, ***$P < 0.001$, ****$P < 0.0001$.

**Data availability.** Coordinates and structure factors for MAG$_{1-5}$ deglycosylated (spacegroup $P3_2 2$), MAG$_{1-5}$ methylated (spacegroup $P6_5 22$), MAG$_{1-3}$ (spacegroup $P1$) and MAG$_{1-3}$ ligand bound (spacegroup $P1$) have been deposited in the Protein Data Bank with succession numbers 5LF5, 5LFU, 5LFR and 5LFV, respectively. All SAXS data is made available at the small angle scattering databank (SASBDB) with the accession codes SASDB55 (glycosylated MAG$_{1-5}$ wt), SASDB26 (glycosylated MAG$_{1-5}$ I473E), SASDB36 (glycosylated MAG$_{1-5}$ N406Q), SASDB46 (glycosylated MAG$_{1-3}$), SASDBF6 (deglycosylated MAG$_{1-5}$ wt) SASDB56 (deglycosylated MAG$_{1-5}$ I473E), SASDB66 (deglycosylated MAG$_{1-5}$ N406Q) and SASDB76 (deglycosylated MAG$_{1-3}$).

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

## Acknowledgements

We thank the staff of the ESRF beamline ID23-1, Swiss Light Source beamline PX and Deutsches Elektronen-Synchrotron beamline PETRA III P14 for assistance with

diffraction data collection and the staff of ESRF beamline BM29 for SAXS measurements. We thank Mike F. Renne and Jonas M. Dörr for assistance in preparing the GT1b liposomes used in the SPR experiment. This work was funded by a Vidi grant (723.012.002) from the Netherlands Organization for Scientific Research (NWO) to B.J.C.J. and an Investment Grant NWO Medium (721.012.004). J.S. and A.J.R.H. are supported by Proteins@Work (project number 184.032.201), and by the Gravity Program Institute for Chemical Immunology, both funded by NWO. S.L. and R.J.P. are funded by an ALW-Vici Grant from NWO, Stichting ParkingsonFonds and Dynamics of Youth Seed Money grant from Utrecht University.

## Author Contributions

M.F.P. and B.J.C.J. designed the experiments. M.F.P. generated constructs, purified proteins and did all the structural biology (SAXS and X-ray diffraction). M.F.P. and D.M.E.T.-W. performed SE-AUC experiments and analysed the data. J.S. performed mass spectrometry experiments and analysed the data together with A.J.R.H.; S.L. performed neurite outgrowth experiments and analysed the data together with R.J.P.; B.J.C.J. supervised the project. M.F.P. and B.J.C.J. wrote the manuscript with input from all authors.

## Additional information

**Competing financial interests:** The authors declare no competing financial interests.

