## [Peer Review File · Nature Communications]

Reviewers' comments:

Reviewer #1 (Remarks to the Author):

A. The manuscript describes the first-ever high-resolution X-ray crystallographic structural studies of the ectodomain of myelin-associated glycoprotein. Additional site-directed mutagenesis, surface plasmon resonance, small-angle X-ray scattering, and ultracentrifugation assays were used to define the nature of dimerization. Neurite outgrowth assays in cell culture demonstrated the importance of dimeric assembly in trans and of carbohydrate recognition on axonal membranes in inhibiting neurite outgrowth.

B. The structures of these domains of this protein have not yet been determined, and were last modelled in the early 1990s (which papers were interesting to re-examine in this new context). The novelty is unequivocal - this study has been 25 years in the making. The protein is an essential one for myelin formation and nervous system integrity, and one that has been identified as an inhibitor of repair upon damage. The study provides a firm molecular foundation upon which to devise therapeutic strategies to overcome this inhibition.

C. The experiments were appropriate and complementary, and well-executed and analyzed. The strategies for mutagenesis were reasoned and insightful.

D. Statistics were fine. In Figure 4, the authors plot the length of the longest neurite. I wonder if it is more meaningful to compare instead the AVERAGE length of, say, a few hundred neurites.

E. Conclusions were appropriate and justified.

F. In addition to the comment on neurite outgrowth assays in part D, I would suggest visible arrows to denote regions of interest, e.g., the stick representations of the carbohydrates. I appreciate that the figures must be full but often these structures are very small to see. In Figure 3f, does the cartoon representation have the correct relative sizes of the protein and the vesicle? This should be stated either way.

In the supporting information, Figures S11-S14 were too condensed and it was hard to glean any meaning from them.

G. Referencing was fine.

H. The paper was well-written and clear, and a pleasure to read.

Reviewer #2 (Remarks to the Author):

A Summary of the key results

The work by Pronker et al. provides a valuable insight into the molecular details of the interactions between myelinating cells and the neuronal membrane. The 3D structure of MAG has remained unknown at high resolution to date, and the authors provide the structure of a dimeric full-length extracellular domain, comprised of 5 Ig domains. The authors have used both X-ray crystallography, biophysical techniques, and a cell culture system to show how dimerization and ganglioside binding by MAG may regulate its interactions and the maintenance of the myelin periplasmic spacing.

B Originality and interest: if not novel, please give references

The structural data are original and valuable. The detailed mechanisms of MAG action have remained elusive, and for example, the long-held belief that Ig4-5 domains interact directly with the neuronal surface is refuted by the current study. The dimensions of MAG directly can explain the 12-14 nm extracellular spacing observed in non-compact myelin.

C Data & methodology: validity of approach, quality of data, quality of presentation

The study has been carried out using mainly structural biology techniques, as appropriate.

The MAG1-3 construct was refined at 2-Å resolution, while the MAG1-5 data extend only to appr. 4 Å. Hence, some discussed features from the lower-resolution data demand further clarification (see below).

The SAXS experiment description should be improved (see below).

D Appropriate use of statistics and treatment of uncertainties

Standard crystallography and SAXS techniques were used, and any remarks are given below. For the neurite outgrowth assay, I am wondering how it was made sure that the same level of the MAG constructs was present on the coverslips to start with.

E Conclusions: robustness, validity, reliability

The conclusions about dimerization and ganglioside binding appear robust, apart from one point of view: the work was done on a truncated protein in vitro, and some conclusions about the effects in vivo during myelination would require further experiments. Thus, I would read through the discussion carefully and make clear when we are reading about in vitro results. Full-length MAG in vivo was not studied here.

F Suggested improvements: experiments, data for possible revision

These are my suggestions for further improvement/correction/consideration:

1. introduction line 47 mentions "misfolding mutations"; has it been shown that misfolding occurs=
2. the myelin-axon spacing of 9-11 nm is cited, while literature in general talks about 12-14 nm. I do not believe the latter was measured from the middle of the membrane, as one figure legend suggests.
3. line 62: MAG is the "oldest member"? Does this mean evolutionarily, or was it first cloned?
4. Electron density maps should be shown for the disulfides between C37-C165 and C421-C430. The latter must be based on 4Å data, so should be unequivocal.
5. Supplementary data could show the SEC behaviour of the constructs, as it is discussed in the text.
6. As far as I see, no chi2 values for the fit of the ab initio SAXS models are given.
7. The concentration dependence of the SAXS results should be discussed.
8. The affinity of the dimer in solution should be compared to the concentrations used in the experiments.
9. Line 321 "rare yet conserved" could be clarified. As far as I understand, the mannosylation is rare and the Trp conserved, but we do not know if the mannosylation is conserved.
10. Lines 340-350; distances of 16 nm are cited - is there evidence for such, or is this just speculation based on the structure?
11. lines 362 and 382, confusing wordings "MAG as a ligand into neurons" and "MAG as a receptor into oligodendrocytes"
12. Methods: crystallography data processing software not mentioned.
13. Line 465 - if Ig 3 has high B factors, could this hint towards a hinge-like motion, which might actually fit with earlier data on the extracellular domain?
14. SAXS methods should give details on the used concentrations and concentration dependence. What data were actually used in the modeling (one concentration, several, extrapolation to zero, ...).
15. Table 1:
I/sI should be <I/sI>
cc(1/2) should be reported

B-factors are average B factors

please report Molprobity overall score and percentile for each structure

16. Table 2. The I(0) column actually gives the MW based on I(0), if we believe the methods section.

17. Fig 2. (d) give the contour level. (g) DAMMIF fit should be shown somewhere.

18. Fig 3. (c) contour level should be given.

19. Fig 5. Yellow sphere and the arrow are invisible at least when printed on paper. Please also see my comment on the 12-14 vs 9-11 nm spacing above.

G References: appropriate credit to previous work?

Referencing seems to be appropriate.

H Clarity and context: lucidity of abstract/summary, appropriateness of abstract, introduction and conclusions

Overall, the paper is clearly written and presented. I am wondering if the discussion could be shortened such that the supplementary discussion could also have its main points in the main text?

Reviewer #3 (Remarks to the Author):

This is the first report of the structure of MAG – a storied neural CAM with roles in maintaining axon-myelin/periaxonal spacing, in potential bi-directional signaling in myelination and axon regeneration (although the importance of the latter in vivo is unclear); MAG is also a founding member of the small family of SgIgLecs. Here, the authors report the structure of MAG's ectodomain based on crystal structures supplemented with modeling. They then carry out a systematic analysis of the predicted structural features seeking to corroborate effects on dimerization, ligand binding, and neurite outgrowth.

The authors report that the MAG's ectodomain has an extended, rigid structure of ~ 10 nM, sufficient to bridge the axon-myelin space – in contrast to earlier reports suggesting a horseshoe structure. They also report MAG functions as a weak cis dimer bridged by hydrophobic interactions of Ig domains 4 and 5, and that it binds to gangliosides in trans via previously implicated R118 and other adjacent sequences in its first Ig domain.

In general, the studies are well done, impressively broad, and the paper is clearly written. The findings provide further insights that will be of interest to investigators in the myelin and

regeneration field.

My only concern is that the crystal structures are based on hypoglycosylated versions of the ectodomain generated in glycosylation-deficient HEK293 cells. Domain 1- 5 are ~ 65 kD (Table 2) vs. the predicted 85 kD for the normally glycosylated ectodomain. Dimerization is enhanced by endo-H deglycosylation and in the N406Q mutant that abolishes a single glycosylation site. Accordingly, it is difficult to fully evaluate the biological significance of the weak dimer that form by their constructs. The authors should comment on whether the position of the other, fully glycosylated sites would be expected to impair cis dimerization further based on their structures due to steric effects or other interactions.

As a minor point: Interpretation of the extent of neurite inhibition depends on equivalent absorption of the various MAG constructs to the coverslip. An ELISA or other assay to evaluate that these constructs adhere equivalently would strengthen this set of findings

Point-by-point response to comments and suggestions on manuscript NCOMMS-16-17733-T.

Adjustments were made to the abstract, figure legend 3 and section headings to comply with the journal standards. All changes are reported as track changes in the revised manuscript.

Reviewers' comments:

Reviewer #1 (Remarks to the Author):

A. The manuscript describes the first-ever high-resolution X-ray crystallographic structural studies of the ectodomain of myelin-associated glycoprotein. Additional site-directed mutagenesis, surface plasmon resonance, small-angle X-ray scattering, and ultracentrifugation assays were used to define the nature of dimerization. Neurite outgrowth assays in cell culture demonstrated the importance of dimeric assembly in trans and of carbohydrate recognition on axonal membranes in inhibiting neurite outgrowth.

B. The structures of these domains of this protein have not yet been determined, and were last modelled in the early 1990s (which papers were interesting to re-examine in this new context). The novelty is unequivocal - this study has been 25 years in the making. The protein is an essential one for myelin formation and nervous system integrity, and one that has been identified as an inhibitor of repair upon damage. The study provides a firm molecular foundation upon which to devise therapeutic strategies to overcome this inhibition.

C. The experiments were appropriate and complementary, and well-executed and analyzed. The strategies for mutagenesis were reasoned and insightful.

Q: D. Statistics were fine. In Figure 4, the authors plot the length of the longest neurite. I wonder if it is more meaningful to compare instead the AVERAGE length of, say, a few hundred neurites.

A: Assessing the longest neurite instead of the average length of the neurites is widely used in the field, see for example Atwal et al., Science 2008 (volume 322, p967-970). We have assessed several hundreds of longest neurites for each condition. Measuring the average length of all the neurites is not expected to provide more information.

E. Conclusions were appropriate and justified.

Q: F. In addition to the comment on neurite outgrowth assays in part D, I would suggest visible arrows to denote regions of interest, e.g., the stick representations of the carbohydrates. I appreciate that the figures must be full but often these structures are very small to see. In Figure 3f, does the cartoon representation have the correct relative sizes of the protein and the vesicle? This should be stated either way.

In the supporting information, Figures S11-S14 were too condensed and it was hard to glean any meaning from them.

A: For clarity we have now added annotated ellipses in Figure 5 to denote regions of interest; i.e. W22 and N406. The cartoon in figure 3f does indeed have the correct relative sizes of the protein and the vesicle. To emphasize this we have added the following to the figure legend “Liposome and MAG molecules are displayed approximately to scale in schematic representation.” We included Supplementary figures S11-S14 (renumbered to S13-16) for completeness to show the quality of the model to the data. All relevant numbers are reported in Table 3. We have not been able to generate less condensed figures with the available software.

G. Referencing was fine.

H. The paper was well-written and clear, and a pleasure to read.

Reviewer #2 (Remarks to the Author):

A Summary of the key results

The work by Pronker et al. provides a valuable insight into the molecular details of the interactions between myelinating cells and the neuronal membrane. The 3D structure of MAG has remained unknown at high resolution to date, and the authors provide the structure of a dimeric full-length extracellular domain, comprised of 5 Ig domains. The authors have used both X-ray crystallography, biophysical techniques, and a cell culture system to show how dimerization and ganglioside binding by MAG may regulate its interactions and the maintenance of the myelin periplasmic spacing.

B Originality and interest: if not novel, please give references

The structural data are original and valuable. The detailed mechanisms of MAG action have remained elusive, and for example, the long-held belief that Ig4-5 domains interact directly with the neuronal surface is refuted by the current study. The dimensions of MAG directly can explain the 12-14 nm extracellular spacing observed in non-compact myelin.

C Data & methodology: validity of approach, quality of data, quality of presentation

The study has been carried out using mainly structural biology techniques, as appropriate. The MAG1-3 construct was refined at 2-Å resolution, while the MAG1-5 data extend only to appr. 4 Å. Hence, some discussed features from the lower-resolution data demand further clarification (see below).

The SAXS experiment description should be improved (see below).

Q: D Appropriate use of statistics and treatment of uncertainties

Standard crystallography and SAXS techniques were used, and any remarks are given below. For the neurite outgrowth assay, I am wondering how it was made sure that the same level of the MAG constructs was present on the coverslips to start with.

A: The adsorption of proteins to poly-L-lysine treated glass coverslips is widely used for cell-based studies. Adsorption of proteins to this surface is nonspecific and it is expected that the MAG

constructs will be immobilized at similar levels. To be able to compare results from different constructs, we use purified protein at well-determined concentrations for adsorption.

Q: E Conclusions: robustness, validity, reliability

The conclusions about dimerization and ganglioside binding appear robust, apart from one point of view: the work was done on a truncated protein in vitro, and some conclusions about the effects in vivo during myelination would require further experiments. Thus, I would read through the discussion carefully and make clear when we are reading about in vitro results. Full-length MAG in vivo was not studied here.

A: To address this we have added the following to the discussion “on the extracellular segment of MAG” as in “...and here by structural and biophysical data on the extracellular segment of MAG indicates that in the periaxonal space...”

F Suggested improvements: experiments, data for possible revision

These are my suggestions for further improvement/correction/consideration:

Q: 1. introduction line 47 mentions "misfolding mutations"; has it been shown that misfolding occurs=

A: For Pelizaeus-Merzbacher disease-like disorder, the S133R mutation was predicted to impair folding of the N-terminal Ig domain based on homology models. Furthermore, it was experimentally shown that this mutant causes MAG to be trapped in the ER and not acquire mature glycosylation, which should have occurred in the Golgi (Lossos et al., Brain, 2015, ref 5). These observations do indicate that MAG S133R has a propensity to misfold. For hereditary spastic paraplegias, the C430G mutation was not experimentally shown to impair folding (Novarino et al., Science, 2015, ref 4). We observe in our crystal structures that the conserved C430 is in a disulfide bond, and from this observation we conclude it is likely that the C430G mutation causes misfolding of MAG. To reflect that the misfolding of this mutation has not been tested experimentally we have changed “for example from misfolding mutations” into “for example from mutations that likely cause misfolding,” in the introduction.

Q: 2. the myelin-axon spacing of 9-11 nm is cited, while literature in general talks about 12-14 nm. I do not believe the latter was measured from the middle of the membrane, as one figure legend suggests.

A: The reviewer raises a valid point. We have reanalyzed the literature and agree with the referee that the myelin-axon spacing in references 10-12 (that we cite in the figure legend on Fig 5) seems not to be measured from the middle of the membranes. However, more recent literature shows that new techniques, based on high-pressure freezing of tissue, have become available that better preserve the ultrastructure of the lipid-rich myelinated axons as compared to chemical fixation reported in references 10-12 (see Möbius et al., Brain Research, 2016, volume 1641, p92-100, for a discussion). In Snaidero et al., Cell, 2014 (ref. 45), high pressure frozen electron micrographs of myelin tissue are analyzed. For example figure 6A shows a high quality electron micrograph, in which the two separate leaflets of both the axon and myelin membranes are visible. The myelin-axon spacing (not including the membranes, thus measured from the cell surfaces) is 9-12 nm in this figure and in agreement with the myelin-axon spacing we determine based on our data. To better reflect the ambiguity that currently exists on the myelin-axon spacing we have removed the “9-11 nm” statements in our manuscript and added the following to the discussion: “This myelin-axon spacing has been reported to be 12-14 nm based on electron micrographs of chemically fixed myelin tissue (refs 10-12). However analysis of more recent electron micrographs of high-pressure frozen myelin that does not suffer from fixation-induced artifacts (ref 45) reveals an axon-myelin spacing of 9-12

nm.” in which reference 45 refers to Snaidero et al., Cell, 2014. In addition we have changed the last sentence of this section to: “The agreement of intermembrane distance determined from high-pressure frozen electron microscopy on myelin tissue (ref 45) and here by structural and biophysical data on the extracellular segment of MAG indicates that in the periaxonal space, MAG is dimerized *in cis* via domains Ig4-Ig5 when bound to axonal gangliosides in trans.” and we have removed the following from figure legend 5: “in close agreement with electron microscopy studies on myelin tissue (refs 10–12, 43). In these studies, a periaxonal diameter of 12-14 nm is reported, measured from the center of the two membranes. This translates to a membrane surface separation of 9-11 nm if a membrane thickness of approximately 3 nm is taken into account”.

Q: 3. line 62: MAG is the "oldest member"? Does this mean evolutionarily, or was it first cloned?

A: This means evolutionarily. To clarify this, we changed “is the oldest member” to “is evolutionarily the oldest member” and added the reference to the end of that sentence (which was before only mentioned at the end of the next sentence).

Q: 4. Electron density maps should be shown for the disulfides between C37-C165 and C421-C430. The latter must be based on 4Å data, so should be unequivocal.

A: The electron density maps for these two non-canonical disulfides are now shown as supplementary figures. We refer to this figure in the text and have renumbered supplementary figures accordingly.

Q: 5. Supplementary data could shown the SEC behaviour of the constructs, as it is discussed in the text.

A: We have now included a supplementary figure (S8) displaying the size exclusion chromatograms of MAG₁₋₅ wt, I473E and N406Q in which the shift to a dimer for N406Q, as referred to in the text, is clearly observable. We refer to this figure in the text and have renumbered supplementary figures accordingly.

Q: 6. As far as I see, no chi2 values for the fit of the ab initio SAXS models are given.

A: We have now included the χ^2 values in the main text and in the figure legend of the supplementary figure with the Dammif fits (supplementary figure 13). In results section “Small Angle X-ray Scattering confirms dimerization interface” we have included the following: “ χ^2 of the model-to-data fit are 1.05 and 1.33 for MAG₁₋₅ I473E and MAG₁₋₅ N406Q, respectively”

Q: 7. The concentration dependence of the SAXS results should be discussed.

A: We have added “Similar concentrations were selected for all samples to allow comparison (see table 2).” to the SAXS methods section. See also our answer to the next question.

Q: 8. The affinity of the dimer in solution should be compared to the concentrations used in the experiments.

A: To address this we have added the following to the paragraph about the affinity in solution (section “The MAG1-5 dimer is weak in solution with a Kd of 3.8×10² μM” in results): “Based on a monomer-dimer equilibrium with these Kd’s, dimer fractions of 18 % for glycosylated wt MAG₁₋₅ and 24 % for deglycosylated wt MAG₁₋₅ are expected to be present in the SAXS experiments (calculated at 52.2 and 37.5 μM for glycosylated and deglycosylated MAG, respectively). Indeed the presence of a mix of monomers and dimers is observed in the SAXS data of both glycosylated and deglycosylated MAG₁₋₅ (Fig. 2J, Supplementary Fig. 11).”

Q: 9. Line 321 "rare yet conserved" could be clarified. As far as I understand, the mannosylation is rare and the Trp conserved, but we do not know if the mannosylation is conserved.

A: Indeed, the mannosylation is rare and the Trp is conserved (as well as the other Trp of the motif), but we do not know if the mannosylation is conserved. To prevent confusion, we changed "rare yet conserved" to "unusual" as the evolutionary conservation is already discussed in the next sentence.

Q: 10. Lines 340-350; distances of 16 nm are cited - is there evidence for such, or is this just speculation based on the structure?

A: There is indeed no direct evidence for a 16 nm distance during compaction of the periaxonal space. Our data indicate a MAG monomer could bridge a 16 nm distance. To clarify this further we have removed "from 16 nm to the final 9-11 nm distance" and added ", based on the length of a MAG monomer" so that it now reads "Additionally, dimerization of MAG could enable compaction of the periaxonal space. The weak *cis*-interaction of MAG, if not stabilized in *trans*, may ensure enough monomer is available to bridge a wider periaxonal spacing (of up to 16 nm, based on the length of a MAG monomer) that may exist during myelin formation."

Q: 11. lines 362 and 382, confusing wordings "MAG as a ligand into neurons" and "MAG as a receptor into oligodendrocytes"

A: To clarify this and prevent confusion we changed "Signaling of MAG as a ligand into neurons can inhibit neurite outgrowth" to "Myelin-to-neuron signaling with MAG as a ligand can inhibit neurite outgrowth" and we changed "Signaling of MAG as a receptor into oligodendrocytes controls myelin formation." to "Axon-to-myelin signaling with MAG as a receptor controls myelin formation."

Q: 12. Methods: crystallography data processing software not mentioned.

A: We have now added the crystallography data processing software including relevant references in the methods section.

Q: 13. Line 465 - if Ig 3 has high B factors, could this hint towards a hinge-like motion, which might actually fit with earlier data on the extracellular domain?

A: This higher B-factor is most likely the result of the fact that in both MAG₁₋₅ structures, Ig3 is the only domain not stabilized by any crystal packing contacts. We think that although there is limited interdomain flexibility, the MAG Ig domains cannot move freely as in beads on a string, based on our SAXS data (Kratky plot, D_{max}, P(r), ab-initio models), as already discussed in our manuscript. To further clarify this, we have added "likely because of the lack of any crystal packing contacts for this domain." to the section discussing the high B-factors of Ig3.

Q: 14. SAXS methods should give details on the used concentrations and concentration dependence. What data were actually used in the modeling (one concentration, several, extrapolation to zero, ...).

A: A single concentration was used for all samples, selecting similar concentrations to allow comparison. To clarify this, we have added the following to the SAXS methods section "Similar concentrations were selected for all samples to allow comparison (see table 2)." and also "A single concentration was used for all measurements, no extrapolation to zero concentration was performed."

Q: 15. Table 1:
I/sI should be

cc(1/2) should be reported

B-factors are average B factors

please report Molprobit overall score and percentile for each structure

A: We have now added $CC_{1/2}$, Molprobit overall scores and percentiles to table 1. "B-factors" is changed to "Average B-factors".

Q: 16. Table 2. The $I(0)$ column actually gives the MW based on $I(0)$, if we believe the methods section.

A: That is correct. To further emphasize this we have changed " I_0 " to " M_m based on I_0 " in Table 2.

Q: 17. Fig 2. (d) give the contour level. (g) DAMMIF fit should be shown somewhere.

A: We have now included the contour level in the figure 2d legend as in " $2F_o - F_c$ electron density at a contour level of 1.0σ ". We have included the DAMMIF fit as a supplementary figure. We refer to this figure in the text and have renumbered supplementary figures accordingly.

Q: 18. Fig 3. (c) contour level should be given.

A: We have now included the contour level in the figure 3c legend as in " $F_o - F_o$ (soaked-unsoaked) electron density at a contour level of 3.0σ "

Q: 19. Fig 5. Yellow sphere and the arrow are invisible at least when printed on paper.

A: Colors were made darker and ellipses with annotation were added to indicate the tryptophan mannosylation on W22 and N-linked glycan at N406.

G References: appropriate credit to previous work?

Referencing seems to be appropriate.

Q: H Clarity and context: lucidity of abstract/summary, appropriateness of abstract, introduction and conclusions

Overall, the paper is clearly written and presented. I am wondering if the discussion could be shortened such that the supplementary discussion could also have its main points in the main text?

A: To address this comment we have added the following to the discussion: "The structure of MAG is the first of a full extracellular portion of a Siglec family member. Besides the common N-terminal V-type Ig domain for recognizing sialic acid moieties, Siglecs vary in the number of additional Ig domains; from 1 up to 16. Comparison of the structures of MAG and Siglec5 reveals a different interdomain orientation between domains Ig1 and Ig2, likely due to differences in amino acids at the interface (Supplementary Fig. 17 and supplementary discussion)". All other sections in the supplementary discussion are already briefly discussed in the results or discussion of the main text.

Reviewer #3 (Remarks to the Author):

This is the first report of the structure of MAG – a storied neural CAM with roles in maintaining axon-myelin/periaxonal spacing, in potential bi-directional signaling in myelination and axon regeneration

(although the importance of the latter in vivo is unclear); MAG is also a founding member of the small family of SigLeCs. Here, the authors report the structure of MAG's ectodomain based on crystal structures supplemented with modeling. They then carry out a systematic analysis of the predicted structural features seeking to corroborate effects on dimerization, ligand binding, and neurite outgrowth.

The authors report that the MAG's ectodomain has an extended, rigid structure of ~ 10 nM, sufficient to bridge the axon-myelin space – in contrast to earlier reports suggesting a horseshoe structure. They also report MAG functions as a weak cis dimer bridged by hydrophobic interactions of Ig domains 4 and 5, and that it binds to gangliosides in trans via previously implicated R118 and other adjacent sequences in its first Ig domain.

In general, the studies are well done, impressively broad, and the paper is clearly written. The findings provide further insights that will be of interest to investigators in the myelin and regeneration field.

Q: My only concern is that the crystal structures are based on hypoglycosylated versions of the ectodomain generated in glycosylation-deficient HEK293 cells. Domain 1- 5 are ~ 65 kD (Table 2) vs. the predicted 85 kD for the normally glycosylated ectodomain. Dimerization is enhanced by endo-H deglycosylation and in the N406Q mutant that abolishes a single glycosylation site. Accordingly, it is difficult to fully evaluate the biological significance of the weak dimer that form by their constructs. The authors should comment on whether the position of the other, fully glycosylated sites would be expected to impair cis dimerization further based on their structures due to steric effects or other interactions.

A: Based on the crystal structures of our full extracellular domains and the location of all (determined and predicted) N-linked glycans in this structure and the structure of myelin-specific N-linked glycans (Sedzik et al., Journal of Neuroscience Research, 2015, volume 93, 1-18), we do not believe that any of the N-linked glycans other than the one on N406 would interfere with dimerization. To clarify this, we added the following: “The other glycans are not expected to interfere with dimerization, also not when considering myelin-specific N-linked glycans³⁹.” to the section “MAG₁₋₅ crystal structures reveal a dimeric arrangement” in the results. Reference 39 refers to Sedzik et al., Journal of Neuroscience Research, 2015.

Q: As a minor point: Interpretation of the extent of neurite inhibition depends on equivalent adsorption of the various MAG constructs to the coverslip. An ELISA or other assay to evaluate that these constructs adhere equivalently would strengthen this set of findings

A: The adsorption of proteins to poly-L-lysine treated glass coverslips is widely used for cell-based studies. Adsorption of proteins to this surface is nonspecific and it is expected that the MAG constructs will be immobilized at similar levels. To be able to compare results from different constructs, we use purified protein at well-determined concentrations for adsorption.

Reviewers' Comments:

Reviewer #1 (Remarks to the Author):

The authors have addressed the reviewers' concerns to my satisfaction and I have no objection to acceptance for publication.

Reviewer #2 (Remarks to the Author):

I have the following comments (marked as NEW COMMENT), related to my earlier concerns and the author responses thereof:

Standard crystallography and SAXS techniques were used, and any remarks are given below. For the neurite outgrowth assay, I am wondering how it was made sure that the same level of the MAG constructs was present on the coverslips to start with. A: The adsorption of proteins to poly-L-lysine treated glass coverslips is widely used for cell-based studies. Adsorption of proteins to this surface is nonspecific and it is expected that the MAG

constructs will be immobilized at similar levels. To be able to compare results from different constructs, we use purified protein at well-determined concentrations for adsorption.

NEW COMMENT: This point was also raised by another referee. It is not enough to simply assume that the different variants immobilize to the same levels.

Q: 4. Electron density maps should be shown for the disulfides between C37-C165 and C421-C430. The latter must be based on 4Å data, so should be unequivocal. A: The electron density maps for these two non-canonical disulfides are now shown as supplementary figures. We refer to this figure in the text and have renumbered supplementary figures accordingly.

NEW COMMENT: I would like to see an omit map for this.

Q: 7. The concentration dependence of the SAXS results should be discussed.

A: We have added “Similar concentrations were selected for all samples to allow comparison (see table 2).” to the SAXS methods section. See also our answer to the next question.

Q: 14. SAXS methods should give details on the used concentrations and concentration dependence. What data were actually used in the modeling (one concentration, several, extrapolation to zero, ...). A: A single concentration was used for all samples, selecting similar concentrations to allow comparison. To clarify this, we have added the following to the SAXS methods section “Similar concentrations were selected for all samples to allow comparison (see

table 2).” and also “A single concentration was used for all measurements, no extrapolation to zero concentration was performed.”

NEW COMMENT: This does not answer the question. How do the variants behave as a function of concentration? It is not enough to measure "similar concentrations".

Q: 15. Table 1: I/sI should be

NEW COMMENT: this was messed up by the submission system, I meant it should be "mean I/sI"

Reviewer #3 (Remarks to the Author):

The authors have been responsive to the concerns raised by each of the reviewers, largely via text clarifications. Overall, this is a significant addition to the field and is a well executed, impressively broad study for which they should be congratulated.

As a side comment: both reviewers 2 & 3 raised the concern that their MAG constructs might adsorb to poly-lysine differently. While it is true that this has not been tested for in neurite outgrowth assays historically - it remains a potential confound. Given the extra work involved for what may be a low yield result, I would not insist it be addressed directly

Point-by-point response to comments and suggestions on manuscript NCOMMS-16-17733-A.

All changes are reported as track changes in the revised manuscript. The additional comments raised by reviewer #2 are addressed below.

Reviewers' comments:

Reviewer 2 (Remarks to the Author):

Standard crystallography and SAXS techniques were used, and any remarks are given below. For the neurite outgrowth assay, I am wondering how it was made sure that the same level of the MAG constructs was present on the coverslips to start with. A: The adsorption of proteins to poly-L-lysine treated glass coverslips is widely used for cell-based studies. Adsorption of proteins to this surface is nonspecific and it is expected that the MAG

constructs will be immobilized at similar levels. To be able to compare results from different constructs, we use purified protein at well-determined concentrations for adsorption.

NEW COMMENT: This point was also raised by another referee. It is not enough to simply assume that the different variants immobilize to the same levels.

A: To clarify that we have not tested the immobilization efficiency in the neurite outgrowth assays, we have added the following sentence to the methods section of the neurite outgrowth assays: "It is expected that proteins immobilize non-specifically to the coverslips, therefore the immobilization efficiency of the different MAG variants was not experimentally verified."

Q: 4. Electron density maps should be shown for the disulfides between C37-C165 and C421-C430. The latter must be based on 4Å data, so should be unequivocal. A: The electron density maps for these two non-canonical disulfides are now shown as supplementary figures. We refer to this figure in the text and have renumbered supplementary figures accordingly.

NEW COMMENT: I would like to see an omit map for this.

A: We have now included omit maps in supplementary figure 3. Torsion angle simulated annealing was performed with Phenix, on a model in which the sulfur atoms in question have been removed, to reduce model bias before calculating the omit maps. The two non-canonical disulfide bonds, between C37-C165 and C421-C430, are clearly visible in the omit maps.

Q: 7. The concentration dependence of the SAXS results should be discussed.

A: We have added "Similar concentrations were selected for all samples to allow comparison (see table 2)." to the SAXS methods section. See also our answer to the next question.

Q: 14. SAXS methods should give details on the used concentrations and concentration dependence. What data were actually used in the modeling (one concentration, several, extrapolation to zero, ...).

A: A single concentration was used for all samples, selecting similar concentrations to allow comparison. To clarify this, we have added the following to the SAXS methods section "Similar concentrations were selected for all samples to allow comparison (see table 2)." and also "A single concentration was used for all measurements, no extrapolation to zero concentration was performed."

NEW COMMENT: This does not answer the question. How do the variants behave as a function of concentration? It is not enough to measure "similar concentrations".

A: As expected for a monomer-dimer equilibrium, SAXS parameters (R_g , D_{max} , M_m based on I_0 etc) are concentration dependent when measuring at concentrations close to the K_D . For all samples analyzed

by SAXS in the paper, three or more concentrations were measured. As expected, for both MAG₁₋₅ wt and for the dimer-enhancing mutant MAG₁₋₅ N406Q (both glycosylated and deglycosylated) we observe concentration dependence in the R_g , M_m based on I_0 , D_{max} and Porod volume. In addition and as expected for the monomeric MAG₁₋₃ and MAG I473E, these parameters were not concentration dependent. To represent the SAXS data in the manuscript we decided to report SAXS data of the samples at concentrations at which sample aggregation and radiation-induced damage were not observed (this occurred often at the high protein concentrations) and clear differences between the mutants are present (see figure 2, table 2 and supplementary figures 10-13). Analyzing this single concentration for each sample provides sufficient information to compare all samples on a qualitative level (see figure 2, table 2 and supplementary figures 10-13). We have used sedimentation equilibrium analytical ultracentrifugation experiments to quantify the monomer-dimer equilibrium (see Table 3).

Q: 15. Table 1: I/sI should be

NEW COMMENT: this was messed up by the submission system, I meant it should be "mean I/sI"

A: To address this point, "I/sI" was replaced with "Mean I/sI" in table 1.